# Learning Dynamics of Zeroth-Order Optimization: A Kernel Perspective

Zhe Li [1]   Bicheng Ying [2]   Zidong Liu [3]   Haibo Yang [1]

## Abstract

Classical optimization theory establishes that zeroth-order (ZO) algorithms suffer from a dimension-dependent slowdown, with convergence rates typically scaling with the model dimension compared to first-order methods. However, in contrast to these theoretical expectations, a growing body of recent work demonstrates the successful application of ZO methods to finetuning Large Language Models (LLMs) with billions of parameters. To explain this paradox, we derive the one-step learning dynamics of ZO SGD, where the empirical Neural Tangent Kernel (eNTK) naturally emerges as the key term governing the learning behavior. Inspection of the eNTK produced by ZO-SGD reveals that each element corresponds to the inner product of neural tangent vectors projected onto a random low-dimensional subspace. Thus, by invoking the Johnson-Lindenstrauss Lemma, our analysis shows that the fidelity of the ZO eNTK is governed primarily by the number of perturbations. Crucially, the approximation error depends on the model output size rather than the massive parameter dimension. This dimension-free property provides a theoretical justification for the scalability of ZO methods to LLMs finetuning tasks. We believe that this kernel-based framework offers a novel perspective for understanding ZO methods within the context of learning dynamics.

## 1. Introduction

Zeroth-order (ZO) optimization (Spall, 2002; Ghadimi & Lan, 2013; Nesterov & Spokoiny, 2017) has recently emerged as a pivotal technique in modern machine learning, particularly in scenarios where gradient computation is computationally prohibitive or strictly unavailable. By estimating gradients solely through function evaluations, ZO methods offer a memory-efficient (Malladi et al., 2023a; Chen et al., 2025) and communication-efficient (Qin et al., 2024; Li et al., 2025; 2026) alternative to first-order (FO) algorithms. These properties have made ZO optimization increasingly popular for deploying large-scale models on resource-constrained edge devices and for black-box adversarial attacks (Chen et al., 2017; Liu et al., 2020).

Despite these practical advantages, ZO methods have historically faced a significant theoretical disadvantage compared to their gradient-based counterparts (Spall, 2002; Conn et al., 2009). Classical optimization theory establishes that ZO algorithms suffer from a dimension-dependent slowdown, with convergence rates typically scaling with the model dimension $d$ (Ghadimi & Lan, 2013; Nesterov & Spokoiny, 2017; Shamir, 2017). In the worst-case scenarios, the variance of the gradient estimator scales linearly with $d$, suggesting that ZO optimization should be prohibitively slow for high-dimensional models (Duchi et al., 2015; Liu et al., 2020). Given that modern deep learning models often possess billions of parameters, this "curse of dimensionality" would theoretically render ZO methods impractical for large-scale training (Golovin et al., 2020; Malladi et al., 2023a).

However, recent empirical breakthroughs contradict this pessimistic theoretical outlook. A growing body of work reports the successful application of ZO methods to finetuning LLMs with billions of parameters (Malladi et al., 2023a; Chen et al., 2024; Zhao et al., 2025). In these high-dimensional tasks, ZO algorithms frequently achieve performance competitive with FO methods, exhibiting convergence behaviors that defy the worst-case scaling laws predicted by classical analysis (Yu et al., 2025). This discrepancy suggests that the standard optimization perspective, which compresses learning dynamics into scalar loss values, fails to capture the structural nuances of how ZO updates drive knowledge acquisition in deep networks (Aghajanyan et al., 2021; Malladi et al., 2023a; Jin & Tan, 2026).

In this work, we resolve this paradox by investigating ZO optimization through the lens of the empirical Neural Tangent Kernel (eNTK) (Jacot et al., 2018). Rather than focusing solely on parameter-space convergence, we analyze the learning dynamics in function space. We show that the effective kernel induced by ZO optimization, which we term the ZO-eNTK, can be understood as a geometric projection of

[1]Department of Computing and Information Sciences Ph.D., Rochester Institute of Technology, Rochester, NY, USA [2]Google Inc., Los Angeles, CA, USA [3]ComboCurve Inc., Houston, TX, USA. Correspondence to: Haibo Yang <hbycis@rit.edu>.

*Proceedings of the 43$^{rd}$ International Conference on Machine Learning*, Seoul, South Korea. PMLR 306, 2026. Copyright 2026 by the author(s).

the standard FO eNTK onto a random subspace spanned by the perturbation vectors. This geometric perspective reveals a fundamental connection between ZO learning dynamics and the Johnson-Lindenstrauss (JL) Lemma (Johnson et al., 1984). By leveraging the JL Lemma, we derive rigorous approximation bounds between the ZO and FO optimization trajectories. Our analysis yields three critical insights:

- Number of Perturbations: The fidelity of the ZO eNTK is mainly governed by the number of perturbations.

- Distribution Robustness: The specific choice of perturbation distribution is secondary to the number of perturbations. Regardless of whether the continuous Gaussian or the discrete Rademacher distribution is used, both are sufficient for effective kernel approximation, explaining the empirical success of simple binary perturbation strategies.

- Dimension Independence: Most importantly, we show that the efficacy of the ZO method depends not on the massive model dimension $d$, but rather on the output vocabulary size $V$. This dimension-free property provides a theoretical justification for the scalability of ZO methods to LLMs, explaining why they remain efficient even when $d$ is vast.

Beyond these specific insights, we believe another contribution of this work lies in establishing a kernel-based framework for ZO optimization, offering a novel perspective that complements the existing optimization-centric view, enriching our understanding of derivative-free learning dynamics.

## 2. Preliminaries of Learning Dynamics and Zeroth-Order Optimization

The standard empirical risk minimization problem in machine learning is typically formulated as

$$\min_{\theta} \ell(\theta) = \sum_{(x,y) \in \mathcal{D}} \mathcal{L}(f_\theta(x), y), \tag{1}$$

where $\theta$ denotes the model parameters, $\ell$ is the objective function, $\mathcal{D}$ represents the dataset containing input-label pairs $(x, y)$, and $\mathcal{L}$ is the loss function (e.g., cross-entropy, mean squared error, or negative log likelihood). From a purely optimization-centric perspective, learning algorithms are often analyzed solely through the objective function $\ell$, which brings considerable analytical simplicity. For example, the ZO (stochastic) gradient descent rule can be concisely written as

$$\theta_{t+1} = \theta_t - \eta \left[ \frac{\ell(\theta_t + \mu u_t) - \ell(\theta_t - \mu u_t)}{2\mu} u_t \right] \tag{2}$$

$$= \theta_t - \eta \langle \nabla \ell(\theta_t), u_t \rangle u_t + \mathcal{O}(\mu \eta), \tag{3}$$

where $t$ is the iteration index, $\eta$ is the learning rate, $\mu > 0$ is the smoothing parameter, and $u \in \mathbb{R}^d$ is the random perturbation vector sampled from a certain distribution (e.g., Gaussian). The estimator $\frac{\ell(\theta_t + \mu u_t) - \ell(\theta_t - \mu u_t)}{2\mu} u_t$ corresponds to the Simultaneous Perturbation Stochastic Approximation (SPSA) gradient with central differences (Spall, 2002; Nesterov & Spokoiny, 2017). Eq. (3), derived via Taylor expansion, provides a more tractable form for analysis. Crucially, this formulation implies that for a sufficiently small smoothing radius $\mu$, the SPSA estimator serves as a proxy for the directional derivative along the random direction $u_t$.

While analytically convenient, an exclusive focus on optimization obscures structural information intrinsic to the learning process of modern machine learning models (Zhang et al., 2017; Belkin et al., 2019). More specifically, as shown in Eq. (2), standard optimization metrics focus exclusively on the objective $\ell(\cdot)$, compressing the complex interplay between the model $f_\theta$ and the data $(x, y)$ into a single scalar loss value. This reduction significantly obscures the structural nuances of the learning trajectory. Thus, it fails to capture how model updates influence predictions at the level of individual data points.

To bridge the gap between optimization and data, we adopt the framework of *learning dynamics*, which characterizes how a model's confidence in the observed data evolves after a training step on the training sample (Ren & Sutherland, 2025). Consider a concrete supervised learning model case:

$$z = h_\theta(x), \quad f_\theta(x) = \texttt{Softmax}(z) \in \mathbb{R}^V,$$

where $h(\cdot)$ denotes the neural network backbone, and $V$ represents the vocabulary size or the number of classes. To quantify the model's confidence in output $y$ given input $x$ at iteration $t$, we define $\pi_{\theta_t}(y|x)$, or concisely $\pi_t(y|x)$. In a multi-class classification setting, this is expressed as:

$$\pi_t(y|x) = \mathbb{1}_y^\top f_\theta(x) = \texttt{Softmax}(h_\theta(x))[y],$$

where $\mathbb{1}_y$ is the one-hot encoding vector with the $y$-th entry equal to 1. For the NLP generation problem, $x$ and $y$ represent a sequence of wordings. We can reformulate the model belief as $\pi_t(y|x) = \prod_{l=1}^L \mathbb{1}_{y_l}^\top f_\theta(x, y_{<l})$, where $y_{<l}$ is the model's previous output sequence smaller than $l$. For clarity, this paper focuses on the multi-class classification case.

Now we are ready to present the learning dynamics for an observed data point $\mathbf{x}_o$, which is defined as the change in log-probability:

$$\Delta \log \pi_t(y|\mathbf{x}_o) := \log \pi_{\theta_{t+1}}(y|\mathbf{x}_o) - \log \pi_{\theta_t}(y|\mathbf{x}_o). \tag{4}$$

Suppose we would like to study how the model's prediction on $\mathbf{x}_o$ changes after one-step update of ZO-SGD using $\mathbf{x}_u$. Applying a FO Taylor expansion yields:

$$\Delta \log \pi_t(y|\mathbf{x}_o)$$

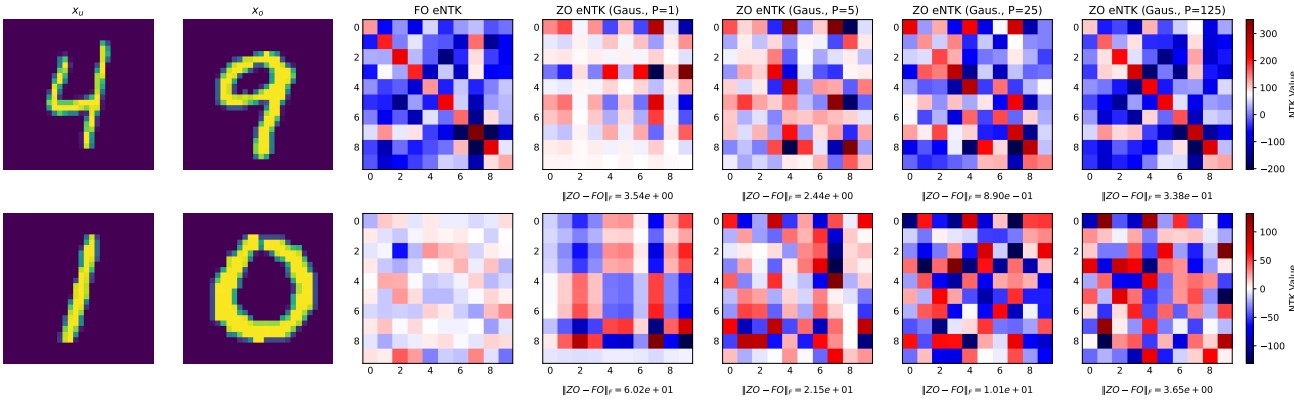

*Figure 1.* ZO eNTK v.s. FO eNTK. The pair with high similarity: 4 and 9. The pair with low similarity: 1 and 0. The relative Frobenius norm error $\|ZO - FO\|_F = \frac{\|\mathcal{K}(\mathbf{x}_u,\mathbf{x}_o) - \mathcal{K}(\mathbf{x}_u,\mathbf{x}_o;U_{t,P})\|_F}{\|\mathcal{K}(\mathbf{x}_u,\mathbf{x}_o)\|_F}$.

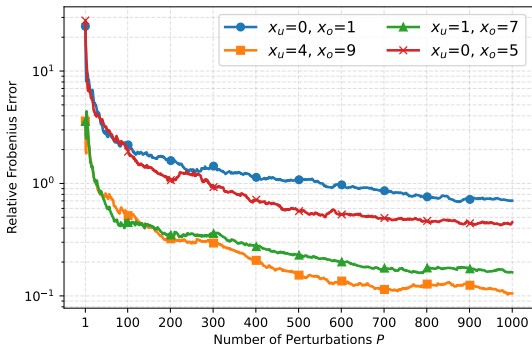

*Figure 2.* Convergence of Frobenius Norm Error between the ZO and FO eNTK. Pairs with high similarity: 4 and 9, 1 and 7. Pairs with low similarity: 0 and 1, 0 and 5.

$$= \log \pi_t(y|\mathbf{x}_o) + \langle \nabla_\theta \log \pi_t(y|\mathbf{x}_o), \theta_{t+1} - \theta_t \rangle$$
$$+ \mathcal{O}(\|\theta_{t+1} - \theta_t\|^2) - \log \pi_t(y|\mathbf{x}_o). \quad (5)$$

For a sufficiently small learning rate, introducing the ZO-SGD update gives[1]

$$\langle \nabla_\theta \log \pi_t(y|\mathbf{x}_o), \theta_{t+1} - \theta_t \rangle$$
$$\approx -\eta [\nabla_\theta \log \pi_t(y|\mathbf{x}_o)] u_t u_t^\mathsf{T} \nabla_\theta \mathcal{L}(f_\theta(\mathbf{x}_u), \mathbf{y}_u)$$
$$= -\eta \underbrace{\nabla_z \log \pi_t(y|\mathbf{x}_o)}_{V \times V} [\underbrace{\nabla_\theta z(\mathbf{x}_o)^\mathsf{T}}_{V \times d} \underbrace{u_t u_t^\mathsf{T}}_{d \times d} \underbrace{\nabla_\theta z(\mathbf{x}_u)}_{d \times V}] \underbrace{\nabla_z \mathcal{L}(z, \mathbf{y}_u)}_{V \times 1}$$
$$= -\eta \mathcal{A}_t(\mathbf{x}_o) \mathcal{K}_t(\mathbf{x}_o, \mathbf{x}_u; u_t) \mathcal{G}_t(\mathbf{x}_u, \mathbf{y}_u), \quad (6)$$

where $\mathcal{A}_t(\mathbf{x}_o) = I - \mathbb{1}\pi_{\theta_t}(\mathbf{x}_o)^\mathsf{T}$ relies only on the model's prediction on the observing data $\mathbf{x}_o$, and $\mathcal{G}_t$ is the last layer's gradient of the previous updating data point. The central quantity in Eq. (6) is $\mathcal{K}_t(\mathbf{x}_o, \mathbf{x}_u; u_t)$, which can be interpreted as an eNTK of the logits $z$, projected onto the random perturbation $u_t$[2] Comparing (6) with the FO learning dy-

---

[1] In some literature, $\nabla_\theta z(\mathbf{x}_u)$ is denoted as a $V \times d$ matrix. Yet, in this paper, we choose the $d \times V$ one due to the later inner product interpolation instead of the outer product one.

[2] To simplify the exposition in (6), we disregard the $\mathcal{O}(\mu\eta)$ smoothing error introduced by the finite difference scheme. This simplification is justified in practice since $\mu$ is generally a small

namics kernel $\mathcal{K}_t(\mathbf{x}_o, \mathbf{x}_u) := \nabla_\theta z(\mathbf{x}_o)^\mathsf{T} \nabla_\theta z(\mathbf{x}_u)$ studied in (Jacot et al., 2018; Ren & Sutherland, 2025), we observe that the key distinction between ZO and FO SGD lies in the presence of the random perturbation $u_t$.

This perspective naturally motivates a kernel-based formulation of ZO optimization via $\mathcal{K}_t(\mathbf{x}_o, \mathbf{x}_u; u_t)$, enabling a systematic analysis of how different design choices affect learning behavior. For clarity, we begin by analyzing the single-perturbation case. However, since practical applications typically employ multi-perturbation strategies, we will subsequently extend our analysis to that setting. Before delving into the theoretical rigor of the general multi-perturbation case, we first examine empirical results to gain intuitive insights from the convergence trajectories.

### 2.1. Example of Learning Dynamics: Zeroth-Order v.s. First-Order SGD.

To empirically evaluate the approximation fidelity of the ZO eNTK, we conduct experiments using the LeNet model ($d = 29,624$) on the MNIST (LeCun, 1998) dataset. From Figures 1 and 2, we observe a consistent, monotonic reduction in approximation error as the number of perturbations $P$ increases. Figure 1 shows the progressive denoising of the ZO estimate because, as $P$ increases from 1 to 125, the kernel begins to recover the distinct block-diagonal structure of the FO eNTK, indicating improved geometric alignment. Moreover, this convergence is corroborated by the Frobenius error trajectories shown in Figure 2. A pivotal finding from this analysis is the significant variance in convergence rates governed by input similarity:

(1) **High Similarity** (e.g., $\mathbf{x}_u = 4$, $\mathbf{x}_o = 9$): Input pairs exhibiting highly semantic similarity show rapid error decay. As visibly evident in the first row of Figure 1, the

---

fixed constant. From a theoretical perspective, one can always select a sufficiently small bound to ensure this term is negligible; alternatively, the Gaussian randomized smoothing approach can be employed for a more refined treatment (see Section 2 of Nesterov & Spokoiny, 2017).

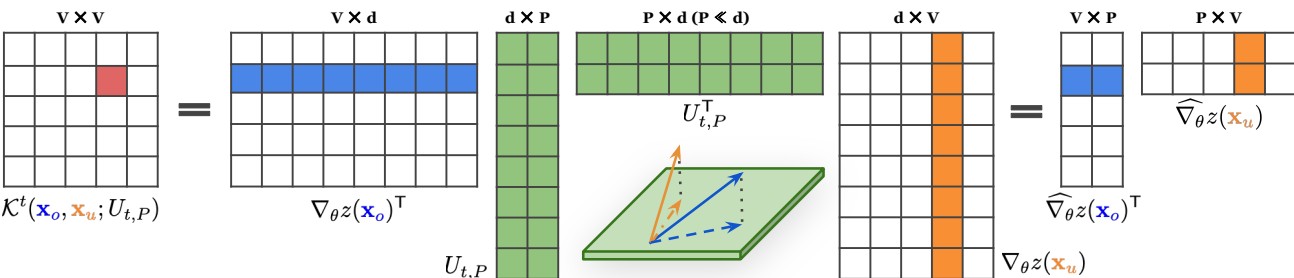

*Figure 3.* Calculating and Interpolating of One Element of Projected Empirical Neural Tangent Kernel.

ZO approximation effectively converges to the ground truth. At $P = 125$, the ZO eNTK visually becomes almost the same as the FO eNTK, achieving a minimal Frobenius norm difference of $\approx 0.338$.

(2) **Low Similarity** (e.g., $\mathbf{x}_u = 0, \mathbf{x}_o = 1$): Structurally distinct pairs suffer from slower convergence and persistent structural discrepancies. Figure 1 shows, even at $P = 125$, the ZO eNTK remains markedly dissimilar to the FO eNTK, failing to fully capture the sharp geometric features of the kernel. This is also clearly reflected in Figure 2, where these pairs with low similarity sustain elevated error margins even as perturbations increase.

Collectively, these results imply that the perturbation sample complexity of ZO gradient estimation is actually highly sensitive to the similarity of $\mathbf{x}_u$ and $\mathbf{x}_o$: the estimator is substantially more perturbation-efficient when evaluating gradients across semantically similar inputs. We provide more results in Figures 8, 9, 10, 11, 12 and 13 in Appendix D.

# 3. Understanding Zeroth-Order Optimization from the Kernel Lens

In this section, we investigate the properties of the projected eNTK induced by ZO methods in greater depth. Specifically, we first extend our analysis (6) to the multi-perturbation case, revealing a basic connection to the Johnson-Lindenstrauss (JL) Lemma (Johnson et al., 1984). By this insight, we examine how the choice of the perturbation distribution $u_t$ and the number of random perturbations $P$ influence the learning dynamics.

### 3.1. Theoretical Foundation

To improve the fidelity of gradient approximation, ZO methods usually employ a multi-perturbation strategy. Formally, the update rule for multi-perturbation ZO-SGD is:

$$\theta_{t+1} = \theta_t - \frac{\eta}{P} \sum_{p=1}^{P} \left[ \frac{\ell(\theta_t + \mu u_{t,p}) - \ell(\theta_t - \mu u_{t,p})}{2\mu} u_{t,p} \right],$$

where $P$ denotes the number of random perturbations sampled at each iteration. By extending the derivation procedure used for the single-perturbation eNTK, we can readily for-

mulate the multi-perturbation projected eNTK as:

$$\mathcal{K}^t(\mathbf{x}_o, \mathbf{x}_u; U_{t,P}) = \underbrace{\nabla_\theta z(\mathbf{x}_o)^\mathsf{T}}_{V \times d} \underbrace{U_{t,P} U_{t,P}^\mathsf{T}}_{d \times d} \underbrace{\nabla_\theta z(\mathbf{x}_u)}_{d \times V}, \quad (7)$$

where $U_{t,P} \in \mathbb{R}^{d \times P}$ represents the random projection matrix induced by the multiple perturbations. Specifically, $U_{t,P}$ is constructed by horizontally stacking the normalized perturbation vectors $\{u_{t,p}/\sqrt{P}\}_{p=1}^{P}$. We visually demonstrate the calculation of projected eNTK $\mathcal{K}(\cdot; U_{t,P})$ in Figure 3.

**Difference between FO and ZO eNTK Update:** Direct analysis of the angle between the ZO gradient estimator $\frac{\ell(\theta_t + \mu u_t) - \ell(\theta_t - \mu u_t)}{2\mu} u_t$ and the exact FO gradient $\nabla \ell(\theta_t)$ reveals that they are nearly orthogonal in high-dimensional spaces when $u_t$ is drawn from a Gaussian or similar isotropic distribution (Vershynin, 2018). However, practical machine learning applications prioritize model output confidence over raw parameter alignment. By characterizing the discrepancy in learning dynamics between FO and ZO methods, we derive the following relationship:

$$\Delta_{\text{fo}} \log \pi_{t+1}(y|\mathbf{x}_o) - \Delta_{\text{zo}} \log \pi_{t+1}(y|\mathbf{x}_o)$$
$$\approx - \eta \mathcal{A}_t(\mathbf{x}_o) \underbrace{[\mathcal{K}_t(\mathbf{x}_o, \mathbf{x}_u) - \mathcal{K}_t(\mathbf{x}_o, \mathbf{x}_u; U_{t,P})]}_{\Delta\mathcal{K}:\text{Kernel Approximation Error}} \mathcal{G}_t(\mathbf{x}_u, \mathbf{y}_u).$$
$$(8)$$

The above quantifies the update discrepancy by mapping it directly to the deviation between the standard eNTK, $\mathcal{K}_t(\cdot)$, and the ZO-induced projected kernel, $\mathcal{K}_t(\cdot; U_{t,P})$. This formulation implies that the fidelity of the ZO update fundamentally depends on how well the random projection matrix $U_{t,P}$ preserves the geometry of the original kernel space. Unlike the raw log-probability difference, the term $\Delta\mathcal{K}$ provides a more structured and tractable metric. Consequently, we utilize this eNTK difference to analyze how the choice of sampling distribution and the number of perturbations influence ZO optimization.

We observe that the $(i, j)$-th entry of the kernel discrepancy matrix $\Delta\mathcal{K}$ takes the following form:

$$\Delta\mathcal{K}[i, j] = \nabla_\theta z_i(\mathbf{x}_o)^\top \nabla_\theta z_j(\mathbf{x}_u)$$
$$- \nabla_\theta z_i(\mathbf{x}_o)^\top U_{t,P} U_{t,P}^\top \nabla_\theta z_j(\mathbf{x}_u), \quad (9)$$

where $\nabla_\theta z_i(\mathbf{x}_o) \in \mathbb{R}^d$ denotes the gradient with respect to

the $i$-th output component (logit). This vector corresponds to the $i$-th column of the Jacobian matrix $\nabla_\theta z(\mathbf{x}_o)$. We define the projected gradient as $\widehat{\nabla}_\theta z_j(\mathbf{x}_u) := U_{t,P}^\top \nabla_\theta z_j(\mathbf{x}_u) \in \mathbb{R}^P$ that represents a linear projection of the original gradient from $\mathbb{R}^d$ into a lower-dimensional subspace $\mathbb{R}^P$. Consequently, the kernel discrepancy matrix can be reformulated to explicitly highlight the difference in inner products between the original and projected spaces:

$$\Delta\mathcal{K}[i,j] = \langle \nabla_\theta z_i(\mathbf{x}_o), \nabla_\theta z_j(\mathbf{x}_u)\rangle - \langle \widehat{\nabla}_\theta z_i(\mathbf{x}_o), \widehat{\nabla}_\theta z_j(\mathbf{x}_u)\rangle.$$

The discrepancy derived above is intimately related to the Johnson-Lindenstrauss (JL) Lemma (Johnson et al., 1984; Dasgupta & Gupta, 2003). Although classically formulated in terms of distance preservation, the JL Lemma naturally extends to inner products via the polarization identity.

---

**Lemma 1** (**Johnson-Lindenstrauss Lemma - Inner Product Version**). *For any set $W$ of $n$ points in a high-dimensional space $\mathbb{R}^d$ and any given arbitrarily small error $\epsilon \in (0,1)$, there exists a linear mapping $f : \mathbb{R}^d \to \mathbb{R}^P$, where the much lower dimension $P = \mathcal{O}(\ln n/\epsilon^2)$, such that the inner product of two points in set $V$ remains similar after applying $f$: $\forall \omega_i, \omega_j \in W$,*

$$|\langle f(\omega_i), f(\omega_j)\rangle - \langle \omega_i, \omega_j\rangle| \leq \frac{\epsilon}{2}(\|\omega_i\|^2 + \|\omega_j\|^2).$$

*When $\|\omega_i\| = \|\omega_j\| = 1$, a more concise expression is*

$$|\langle f(\omega_i), f(\omega_j)\rangle - \langle \omega_i, \omega_j\rangle| \leq \epsilon.$$

*The proof of this lemma is provided in Appendix B.*

---

A fundamental and surprising consequence of the JL Lemma is that **the sufficient projection dimension $P$ is independent of the original feature dimension $d$, depending solely on the number of points $n$ and the error tolerance $\epsilon$.** In the context of our ZO and FO comparison, this implies that once the number of perturbations $P$ exceeds a threshold determined by the vocabulary size $V$, the kernel discrepancy remains tightly bounded, regardless of how large the model dimension $d$ becomes. Consequently, a large enough $P$ ensures that the projected kernel $\mathcal{K}_t(\cdot; U_{t,P})$ converges to the true eNTK, thereby aligning the ZO learning trajectory with the ideal FO dynamics and theoretically validating that multi-perturbation strategies minimize the variance of the model's confidence updates.

Quantitatively, the approximation error $\epsilon$ decays at an inverse square root rate with respect to the number of perturbations, specifically $\epsilon = \mathcal{O}(\sqrt{\ln V/P})$. This theoretical scaling aligns with by convergence curve illustrated in Figure 2. Building on this foundation, we next apply the JL framework to analyze two concrete examples, investigating how different projection distributions and the number of perturbations influence the performance of ZO methods.

## 3.2. Impact of Projection Distributions

The perturbation vector $u$ is typically sampled from simple, symmetric distributions to estimate gradient information (Flaxman et al., 2005; Nesterov & Spokoiny, 2017). We consider two commonly choices in the literature

$$\text{Gaussian: } u \sim \mathcal{N}(\mathbf{0}, \mathbf{I}),$$
$$\text{Rademacher: } u \sim \text{Rad}(0.5), \tag{10}$$

where Rademacher distribution is just a discrete distribution with equal probability choosing $\pm 1$.

### 3.2.1. OPTIMIZATION POINT OF VIEW

A substantial body of literature (Sadegh & Spall, 2002; Maheswaranathan et al., 2019; Gao & Sener, 2022; Bollapragada et al., 2024; Sawada et al., 2025; Ma & Huang, 2025; Tan et al., 2025) has investigated how different perturbation distributions impact the performance of ZO methods. These studies primarily focus on optimization or statistical efficiency, such as variance reduction or Mean-Squared Error (MSE) minimization (Gao & Sener, 2022). While a comprehensive analysis of distribution effects in MSE perspective is beyond the scope of this paper, we provide an illustrative derivation of the gradient estimator's second moment, $\mathbb{E}\|\langle\nabla\ell(\theta), u\rangle u\|^2$, for self-containment. This quantity is a critical proxy for convergence speed. For simplicity, we restrict our analysis to the single-perturbation case; note that for multiple perturbations, this error term roughly scales inversely with $P$ due to the independence of samples. In the standard Gaussian case, the expected squared norm of the estimator is derived as follows:

$$\mathbb{E}\|\langle\nabla\ell(\theta), u\rangle u\|^2 = \text{Tr}\left(\mathbb{E}\, uu^\top \nabla\ell(\theta)\nabla\ell(\theta)^\top uu^\top\right)$$
$$\overset{(a)}{=} (d+2)\|\nabla\ell(\theta)\|^2. \tag{11}$$

We provide the detailed algebraic step (a) above, which relies on the fourth-order moments of the Gaussian distribution, in Appendix C. In the Rademacher distribution case, we have something similar:

$$\mathbb{E}\|\langle\nabla\ell(\theta), u\rangle u\|^2 = \mathbb{E}\langle\nabla\ell(\theta), u\rangle^2 \|u\|^2$$
$$\overset{(a)}{=} d\sum_{i,j}\nabla_i\ell(\theta)\nabla_j\ell(\theta)\mathbb{E}_{i,j}u_i u_j$$
$$= d\|\nabla\ell(\theta)\|^2, \tag{12}$$

where $\nabla_i\ell(\theta)$ denotes the $i$-th component of the gradient $\nabla\ell(\theta)$. Step (a) leverages the property that for Rademacher vectors, the squared norm $\|u\|^2 = d$ holds deterministically, which significantly simplifies the derivation compared to the Gaussian case. Finally, the derivation relies on the independence of perturbation entries $\mathbb{E}[u_i u_j] = \delta_{i,j}$.

Above analysis is far from the complete optimization proof. However, it still can shed some intuitions. From this perspective, the Rademacher distribution yields a strictly lower second moment than the Gaussian distribution, though the

relative performance gap diminishes asymptotically as the dimension $d \to \infty$. Nevertheless, regardless of the distribution chosen, the variance of the zeroth-order estimator scales linearly with $d$, making single-perturbation estimators highly inefficient when the model dimension $d$ is vast – a common belief in the optimization point of view.

### 3.2.2. eNTK POINT OF VIEW

The universality of the JL Lemma stems from the concentration of measure phenomenon in high-dimensional spaces. While the generic bound suggests $P = \mathcal{O}(\ln n/\epsilon^2)$, the hidden constant factors are governed by the tail properties of the distribution used to populate the projection matrix $U_{t,P}$. Hence, we can use this to study the impact of distribution.

Formally, for a unit vector $x \in \mathbb{R}^d$ and a scaled projection map $f(x) = U_{t,P}^{\mathsf{T}} x$, the validity of the JL Lemma relies on a tail bound of the form: $\mathbb{P}\left(\left|\|f(x)\|^2 - 1\right| \geq \epsilon\right) \leq 2\exp\left(-c(\mathcal{Q}) \cdot P \cdot \epsilon^2\right)$, where $c(\mathcal{Q})$ is the *concentration constant* specific to the distribution $\mathcal{Q}$. To satisfy the union bound over $n$ points with probability $1 - \delta$, the required projection dimension is:

$$P \geq \frac{2\ln(n) + \ln(1/\delta)}{c(\mathcal{Q}) \cdot \epsilon^2}. \tag{13}$$

A larger concentration constant $c(\mathcal{Q})$ implies a more efficient projection, requiring fewer perturbations $P$ to achieve the same error tolerance $\epsilon$.

**Gaussian Distribution**: The squared norm of the projected vector, $Y = \|U_{t,P}x\|^2$, follows a Chi-Squared distribution with $P$ degrees of freedom ($\chi_p^2$). Applying the Chernoff-Cramér bound, we obtain:

$$\mathbb{P}(Y \geq p(1+\epsilon)) \leq \exp\left(-\frac{p}{2}\left(\epsilon - \ln(1+\epsilon)\right)\right). \tag{14}$$

Using taylor expansion on the $\ln$ term that $\ln(1+\epsilon) = \epsilon - \frac{\epsilon}{2} + \mathcal{O}(\epsilon^3)$, we obtain $c(\mathcal{N}) \approx 1/4$. We can get a similar result for the lower bound $\mathbb{P}(Y \leq p(1-\epsilon))$.

**Rademacher Distribution**: For a fixed unit vector $x$, the projection component $y_k = \sum_{j=1}^d u_{kj}x_j$ is a weighted sum of bounded independent variables. By Hoeffding's inequality, the sub-Gaussian proxy variance is minimal. As established by Achlioptas (2003), the strict boundedness leads to sharper concentration behavior:

$$\mathbb{P}\left(\left|\|f(x)\|^2 - 1\right| \geq \epsilon\right) \leq 2\exp\left(-\frac{p\epsilon^2}{4} + \frac{p\epsilon^3}{6}\right).$$

This derivation yields a concentration constant $c(\mathcal{R}) \approx 1/4$, which is comparable to the standard Gaussian baseline. While this parity aligns with our earlier optimization-based analysis, a key distinction emerges: unlike the variance of the gradient estimator, the concentration bound here is independent of the model dimension $d$. We will further elucidate the implications of this dimension-independent scaling in the subsequent subsection.

This echoes the observation by Achlioptas (2003) that simple discrete projections can achieve performance comparable to complex continuous distributions, which is corroborated empirically in our observation in Figure 4. In Figure 4, we show the comparison of the continuous Gaussian distribution against the discrete Rademacher distribution. To rigorously quantify the approximation quality, the analysis utilizes three complementary metrics: (1) Frobenius Norm Error, which measures the element-wise Euclidean distance between the ZO and FO eNTK matrices; (2) Centered Kernel Alignment (CKA) Error, which evaluates the structural dissimilarity between the representational geometries; and (3) Spectral Distance (Wasserstein distance), which assesses the discrepancy between the eigenvalue distributions of the two kernels. The results demonstrate that as the number of perturbations $P$ increases, both distributions yield a consistent and monotonic reduction in approximation error across all three metrics. A pivotal observation from these trajectories is the distribution robustness of the ZO learning dynamics, as the convergence curves for Gaussian and Rademacher sampling are nearly indistinguishable. This empirical parity is supported by the kernel-based framework and the Johnson-Lindenstrauss (JL) Lemma, which suggest that the concentration constants for both distributions are comparable. Ultimately, Figure 4 confirms that the fidelity of the ZO updates is primarily governed by the perturbation count $P$ rather than the specific sampling distribution.

### 3.3. The Impact of The Number of Perturbations

Besides the sampling distribution, the number of perturbations is another key hyper-parameter in the ZO method. Existing studies (Ghadimi & Lan, 2013; Liu et al., 2018; 2020; Zhang et al., 2024; Li et al., 2025; Peng et al., 2025; Li et al., 2026; Chen & Ma, 2025) have established that increasing the number of perturbations in ZO optimization can improve gradient approximation and accelerate convergence from an optimization-theoretic perspective. These works typically characterize the benefit of using multiple perturbations through variance reduction or improved convergence rates, and thus provide theoretical justification for employing larger perturbation budgets during training. However, a fundamental question remains largely unexplored: *how many perturbations are sufficient to achieve learning dynamics comparable to first-order optimization?* In other words, beyond asymptotic convergence guarantees, it is unclear at what point the ZO updates become a sufficiently accurate approximation of their FO counterparts in practice. In this subsection, we address this question through a kernel-based analysis. By examining how the number of perturbations affects the eNTK induced by ZO updates, we provide a prin-

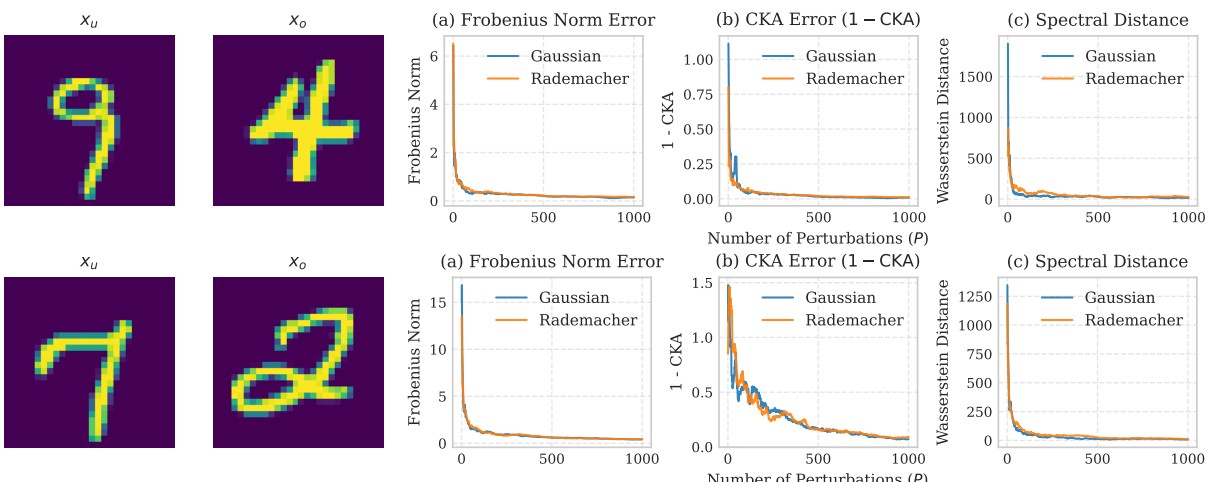

*Figure 4.* Gaussian v.s. Rademacher Distributions (LeNet + MNIST)

cipled criterion for determining when the resulting learning dynamics closely match those of FO optimization.

### 3.3.1. OPTIMIZATION POINT OF VIEW

From the optimization point of view, we just need to examine the ZO-SGD update (2) and the properties of the loss function $\ell$.

Suppose the loss function $\ell(\theta)$ is $L$-smooth and the gradient associated with stochastic function value oracle is unbiased with variance bounded by $\sigma^2$. When the learning rate satisfies $\eta \leq \frac{P}{Ld}$, the ZO-SGD algorithm satisfies the following one-step descent bound:

$$\mathbb{E}\|\nabla\ell(\theta_t)\|^2 \leq \frac{\mathbb{E}\,\ell(\theta_t) - \mathbb{E}\,\ell(\theta_{t+1})}{\eta} + \frac{\eta dL}{2P}\sigma^2 + O(\eta\mu)$$

For a sufficiently large number of iterations $T$, setting the learning rate $\eta = \mathcal{O}\left(\sqrt{\frac{P}{dLT}}\right)$ yields the convergence rate:

$$\frac{1}{T}\sum_{t=0}^{T-1}\mathbb{E}\|\nabla\ell(\theta_t)\|^2 = \mathcal{O}\left(\sqrt{\frac{dL}{PT}}\right). \quad (15)$$

The detailed proof is provided in Appendix C. We highlight two key implications of this convergence rate below. First, the convergence rate improves with the number of perturbations, scaling as $\mathcal{O}(1/\sqrt{P})$, which aligns with what we predicted from JL lemma. Second, the convergence rate scales with the ambient model dimension $d$. Since $d$ is typically vast in modern deep learning models, this dependency underpins the conventional wisdom that zeroth-order methods are significantly slower and less sample-efficient than FO methods in high-dimensional settings.

Recently, Malladi et al. (2023b) proposed the low-effective rank assumption to improve the convergence rate. This hypothesis builds on prior observations that the Hessian

spectrum of a well-trained model is dominated by a few significant eigenvalues (Papyan, 2020; Yao et al., 2020). However, for LLMs, explicitly computing the effective rank is computationally prohibitive. Consequently, it remains difficult to empirically calculate or verify if the low effective rank is truly independent of the model dimension. See more details in Appendix C.2

### 3.3.2. ENTK POINT OF VIEW

We proceed to establish a theoretical upper bound for the discrepancy in model belief updates between the FO and ZO methods. Starting from the operator norm inequality, we have:

$$\|\Delta_{\text{fo}}\log\pi_{t+1}(y|\mathbf{x}_o) - \Delta_{\text{zo}}\log\pi_{t+1}(y|\mathbf{x}_o)\|_2$$
$$\leq \eta\|A\|_2\|\Delta\mathcal{K}\|_F\|\mathcal{G}_t(\mathbf{x}_u, \mathbf{y}_u)\|_2. \quad (16)$$

The critical term is the Frobenius norm of the kernel discrepancy matrix, $\|\Delta\mathcal{K}\|_F$. We bound this term by leveraging the inner-product preservation property of the Johnson-Lindenstrauss Lemma 1. Substituting it into the definition of the squared Frobenius norm, we obtain:

$$\|\Delta\mathcal{K}\|_F^2 \leq \frac{\epsilon^2}{4}\sum_{i=1}^{V}\sum_{j=1}^{V}\left(\|\nabla_\theta z_i(\mathbf{x}_o)\|_2^2 + \|\nabla_\theta z_j(\mathbf{x}_u)\|_2^2\right)^2$$
$$\overset{(a)}{\leq} \frac{\epsilon^2}{2}\sum_{i=1}^{V}\sum_{j=1}^{V}\left(\|\nabla_\theta z_i(\mathbf{x}_o)\|_2^4 + \|\nabla_\theta z_j(\mathbf{x}_u)\|_2^4\right)$$
$$\overset{(b)}{\leq} \frac{\epsilon^2 V}{2}\left(\|\nabla_\theta z(\mathbf{x}_o)\|_F^4 + \|\nabla_\theta z(\mathbf{x}_u)\|_F^4\right)$$
$$\leq \frac{\epsilon^2 V}{2}\left(\|\nabla_\theta z(\mathbf{x}_o)\|_F^2 + \|\nabla_\theta z(\mathbf{x}_u)\|_F^2\right)^2. \quad (17)$$

In step (a), we apply Jensen's inequality $(a + b)^2 \leq 2a^2 + 2b^2$. In step (b), we utilize the property that the sum of squares is bounded by the square of the sum for non-negative terms (i.e., $\sum x_i^2 \leq (\sum x_i)^2$), allowing us to bound the sum

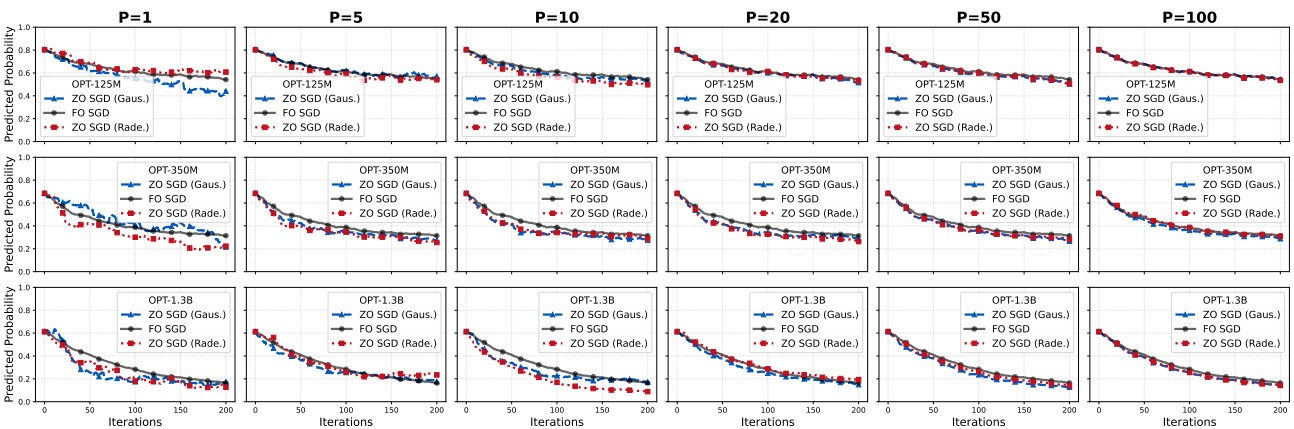

*Figure 5.* ZO Trajectory Comparison between OPT-125M to OPT-1.3B model on SST-2 Task over Different Perturbations.

of fourth powers by the square of the Frobenius norm.

Denote the Frobenius norms of the Jacobian matrix as $\Xi_t = \max(\|\nabla_\theta z(\mathbf{x}_o)\|_F^2, \|\nabla_\theta z(\mathbf{x}_u)\|_F^2)$; Then we have the following succinct formula $\|\Delta\mathcal{K}\|_F \leq \epsilon \cdot \Xi_t \cdot \sqrt{V}$. Substituting this back into Eq. (16) and recalling that $\epsilon \sim \sqrt{(\log V)/P}$, we arrive at the final convergence rate:

$$\|\text{Diff}\|_2 \lesssim \underbrace{\sqrt{\frac{V \log V}{P}}}_{\text{JL Scaling}} \cdot \underbrace{\eta\Xi_t\|\mathcal{G}_t\|_2\|\mathcal{A}_t\|_2}_{\text{Model Dynamics}}. \quad (18)$$

This result indicates that maintaining trajectory fidelity in tasks with large vocabulary sizes $V$ requires the number of perturbations $P$ to scale appropriately with $V$. Typically, the learning rate $\eta$ is chosen to be sufficiently small such that the term $\eta\Xi_t\|\mathcal{G}_t\|_2$ remains bounded, thereby preventing algorithmic divergence. Furthermore, it is straightforward to show that the product $\|\mathcal{A}_t\|_2 \leq 1$ is inherently bounded.

Thus, we arrive at a striking conclusion: the ZO method can effectively approximate the FO trajectory provided the number of perturbations is sufficiently large. Crucially, this requirement depends on the output dimension $V$ rather than the model dimension $d$. This finding offers an alternative explanation for the modest convergence degradation of ZO methods observed empirically: ZO methods can achieve a dimension-independent convergence rate even without the strict low-effective-rank assumption on the Hessian.

### 3.4. Summary and Validation

We conclude this section by synthesizing the insights derived from both the optimization and kernel perspectives and validating it in LLMs tasks. While distinct in their mathematical formulations, these two frameworks offer complementary rather than conflicting views on ZO dynamics.

Both perspectives agree on the fundamental role of the number of perturbations $P$. Whether minimizing the gradient variance (Optimization view) or bounding the kernel ap-

proximation error via the Johnson-Lindenstrauss Lemma (eNTK view), the convergence rate consistently improves with $\mathcal{O}(1/\sqrt{P})$. This confirms that increasing the number of perturbations is a universal strategy for enhancing the fidelity of ZO updates.

However, the crucial distinction lies in how each framework treats dimensionality. The classical optimization perspective typically yields bounds dependent on the massive model dimension $d$, suggesting that ZO methods are prohibitively slow for large-scale models. In contrast, our eNTK analysis reveals a more optimistic reality: the learning trajectory's fidelity depends primarily on the output vocabulary size $V$. This dimension-free property of the projected kernel offers a rigorous theoretical justification for the empirical success of ZO methods in fine-tuning LLM, where $d$ is vast but $V$ is comparatively moderate.

**Validation on LLMs (Trajectory versus model size):** In Figure 5, we empirically validate this theoretical scaling law by examining the learning trajectories of OPT (Zhang et al., 2022) models ranging from 125M, 350M to 1.3B parameters on the SST-2 (Socher et al., 2013; Wang et al., 2018) task. We observe a consistent pattern: as $P$ increases from 1 to 100, the ZO trajectories (for both Gaussian and Rademacher distributions) progressively align with the FO baseline. Crucially, this alignment behavior exhibits remarkable invariance to model scale. Despite the model dimension $d$ increasing by an order of magnitude (from 125M to 1.3B), the threshold of $P$ required to recover the FO dynamics remains virtually constant. For instance, at $P = 50$, ZO-SGD effectively mimics the FO trajectory across all model sizes. This empirical evidence strongly corroborates our kernel-based derivation: the fidelity of ZO learning dynamics is governed by the output dimension $V$ (which remains constant across these models), rather than being diluted by the massive expansion of $d$.

**Validation on LLMs (Difference of trajectory versus**

**output dimensions):** To explicitly validate our theoretical bound, specifically that ZO fidelity degrades at a rate of $\mathcal{O}(\sqrt{V \log V / P})$ without the confounding architectural variables of an LLM, we set up a controlled synthetic experiment. We constructed a two-layer MLP with a fixed hidden dimension and a fixed number of perturbations ($P = 50$). We scaled $V$ from 2 to 1000 and measured the Relative Frobenius Error between FO eNTK and ZO eNTK. As shown in Table 1, the approximation error increases at a rate consistent with $\sqrt{V}$, exactly as predicted by our theorem.

## 4. Trade-Offs, Limitations, and Future Work

As an explanatory study, this work primarily investigates the theoretical and empirical roles of the perturbation budget $P$ in ZO optimization under the standard empirical risk minimization (ERM) setting. Our analysis is grounded in supervised fine-tuning (SFT) tasks within the lazy training regime, where model features remain approximately stable and the empirical NTK varies slowly. Within this scope, our kernel-based analysis confirms that increasing $P$ enhances trajectory fidelity. However, this improvement comes with significant engineering trade-offs and domain-specific limitations that warrant further discussion.

- **Computational Trade-Offs and Parallelization.** Implementing multiple perturbations introduces a linear increase in function query costs, which can be prohibitive for large-scale models. A natural question arises: *can we parallelize these multiple perturbations akin to mini-batch processing to amortize the cost?* While conceptually similar to data parallelism, parallelizing ZO perturbations is non-trivial due to the memory constraints of GPU accelerators. Unlike standard gradient accumulation, evaluating $P$ perturbed forward passes in parallel effectively multiplies the batch size by $P$. For LLMs that already operate near the limit of device memory, this naive parallelization creates a severe bottleneck, risking out-of-memory errors. Future work is needed to investigate efficient activation-free parallelization strategies or hybrid schemes that balance query fidelity with memory overhead.

- **Sequential Dependencies and Task Domains.** Our current eNTK framework is tailored to standard SFT

*Table 1.* ZO eNTK Approximation Error vs. Output Size. Difference norm $k := \|\Delta\mathcal{K}\|_F$, where $\Delta\mathcal{K}$ is defined in (8) and FO Norm $f = \|\mathcal{K}\|_F$ as introduced in Section 2.

| Output Size $V$ | Difference Norm $k$ | FO Norm $f$ | Relative Error |
|---|---|---|---|
| 2 | 21.2548 | 76.8934 | 0.2764 |
| 10 | 81.5816 | 184.9210 | 0.4412 |
| 100 | 957.8461 | 653.4963 | 1.4657 |
| 500 | 5083.9067 | 1706.7668 | 2.9787 |
| 1000 | 6830.7114 | 1849.7047 | 3.6929 |

tasks such as classification, sentiment analysis, and short-form generation (e.g., prompt-based tuning with verbalizers), where the effective output dimension is small and the kernel remains stable. However, it does not cover tasks with strong sequential dependencies, such as long-horizon chain-of-thought reasoning. In such autoregressive settings, outputs at step $t$ influence inputs at step $t+1$, leading to compounding distribution shifts that are not captured by our one-step kernel analysis. Moreover, the effective output dimension grows with sequence length, which may require a significantly larger perturbation budget or a fundamentally different analysis. Extending the projected kernel perspective to capture these temporal dependencies remains an important direction for future work.

- **Dynamics of Pre-training vs. Fine-tuning.** Our analysis relies on the relative stability of the eNTK, which typically holds in fine-tuning scenarios where parameters remain close to their initialization. This assumption enables the random projection induced by ZO to approximate the gradient geometry effectively. However, it breaks down in training-from-scratch settings, where feature learning dominates and the kernel evolves rapidly. Under such non-stationary dynamics, the random subspace projection may fail to track the rapidly changing true gradient direction. This provides a theoretical explanation for the limited success of ZO methods in large-scale pre-training, suggesting that ZO is particularly well-suited for the stable, low-rank adaptation regime characteristic of SFT.

## 5. Conclusion

In this work, we present a novel framework analyzing ZO optimization through the lens of learning dynamics and the eNTK. While classical optimization theory provides essential convergence bounds, our approach offers a complementary perspective that reveals the structural properties of ZO dynamics. By characterizing ZO training as a geometric projection process, we leverage the JL Lemma to prove that the trajectory approximation error is bounded by the number of perturbations $P$ and output vocabulary size $V$, independent of the model dimension $d$. This dimension-free property provides an alternative justification for the scalability of ZO methods to LLM. We believe these insights offer a principled foundation for understanding and improving derivative-free fine-tuning in high-dimensional settings.

## Acknowledgements

This work is supported in part by RIT CHAI Faculty Seed Grant, NVIDIA Academic Grant Program, NIH awards R16GM159671, and NSF grant CNS-2112471. The content is solely the responsibility of the authors and does not nec-

essarily represent the official views of the funding agencies.

## Impact Statement

This paper focuses on the theoretical analysis of the learning dynamics of zeroth-order optimization. As a foundational study aimed at advancing machine learning theory, we do not believe that there are specific negative societal consequences that must be highlighted here.

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

# Appendix

## A. More Related Works

**Neural Tangent Kernel (NTK):** The NTK framework, originally introduced to characterize the training dynamics of over-parameterized neural networks, establishes that in the infinite-width limit, gradient descent optimization is equivalent to kernel regression with a static kernel (Jacot et al., 2018). While this lazy training regime provides strong theoretical guarantees (Arora et al., 2019), subsequent research has focused on the finite-width setting, where the kernel evolves during training, capturing the intricate feature learning process of the network (Hanin & Nica, 2020; Yang & Hu, 2021). Understanding this kernel evolution is essential for accurately describing the learning dynamics of practical deep learning models. However, the application of NTK theory has been predominantly confined to FO optimization methods. In the realm of ZO optimization, the literature has extensively analyzed perturbation distributions, such as Gaussian or uniform smoothing, strictly through the lens of estimator efficiency. Recent works have developed unified frameworks to compare these distributions based on bias, variance, and dimension dependence (Liu et al., 2020; Bollapragada et al., 2024), and have derived optimal distributions to minimize the Mean Squared Error (MSE) of gradient estimators within the parameter space (Gao & Sener, 2022). Recent attempts, such as ZOPO (Hu et al., 2024) and KerZOO (Mi et al., 2025), have begun to incorporate kernel perspectives into derivative-free optimization to improve query efficiency. Crucially, however, the intersection of ZO optimization and NTK dynamics remains unexplored.

**Zeroth-Order (ZO) Optimization:** ZO optimization has been extensively studied in the literature, primarily from a theoretical optimization perspective. Early works focused on convergence guarantees for smooth or stochastic objectives (Flaxman et al., 2005; Nesterov & Spokoiny, 2017), while more recent studies explored the impact of different perturbation distributions on gradient estimation accuracy and optimization efficiency. These efforts have provided valuable insights into variance reduction, estimator bias, and function-value convergence (Liu et al., 2018; Li et al., 2018; Gao & Sener, 2022; Bollapragada et al., 2024). Despite these advances, existing studies largely concentrate on optimization performance and provide limited understanding of the underlying training dynamics induced by ZO methods. In particular, how random perturbations interact with parameter updates and affect the evolution of model representations during training remains largely unexplored. Addressing this gap is crucial for a deeper theoretical and empirical understanding of ZO optimization in modern machine learning and deep learning applications. To reconcile the discrepancy between the empirical success of ZO fine-tuning and its pessimistic worst-case theoretical analysis, Malladi et al. (2023b) proposed the low-effective rank assumption. This hypothesis builds on prior observations that the Hessian spectrum of a well-trained model is dominated by a few significant eigenvalues, while the remaining eigenvalues cluster near zero (Papyan, 2020; Yao et al., 2020) - a condition distinct from being strictly low-rank. Yet, for LLMs, explicitly computing the effective rank is computationally prohibitive. Thus, it remains difficult to empirically verify whether this value is truly independent of the model dimension.

## B. Proof of Lemma 1

First of all, recall the polarization identity that connects the inner product and distances:

$$\langle u, v \rangle = \frac{1}{4}(\|u + v\|^2 - \|u - v\|^2). \tag{19}$$

For any linear function $f$, we also have

$$\langle f(u), f(v) \rangle = \frac{1}{4}(\|f(u + v)\|^2 - \|f(u - v)\|^2). \tag{20}$$

Hence, we just need to show the norms of all vectors in the set $W' = \{u + v | u, v \in W\} \bigcup \{u - v | u, v \in W\}$ are all $\epsilon$-preserved. This set contains at most $2n^2 + n$ elements. Leveraging the original JL lemma, we know there exists a linear mapping function $f : \mathbb{R}^d \to \mathbb{R}^P$, where $P = \mathcal{O}(\ln(|W'|/\epsilon^2) = \mathcal{O}(\ln(n)/\epsilon^2)$, such that for any $w \in W'$ we have

$$(1 - \epsilon)\|w\|^2 \leq \|f(w)\|^2 \leq (1 + \epsilon)\|w\|^2. \tag{21}$$

Substituting the above result into the polarization identity

$$\begin{aligned}
\langle f(u), f(v) \rangle &\leq \frac{1}{4}\left[(1 + \epsilon)\|u + v\|^2 - (1 - \epsilon)\|u - v\|^2\right] \\
&= \frac{1}{4}\left[(\|u + v\|^2 - \|u - v\|^2) + \frac{\epsilon}{2}(\|u + v\|^2 + \|u - v\|^2)\right] \\
&\leq \langle u, v \rangle + \frac{\epsilon}{2}(\|u\|^2 + \|v\|^2).
\end{aligned} \tag{22}$$

Similarly, we can establish the lower bound:

$$\langle f(u), f(v) \rangle \geq \langle u, v \rangle - \frac{\epsilon}{2}(\|u\|^2 + \|v\|^2). \tag{23}$$

Combining the above two results, we complete the proof of this Lemma. $\square$

## C. Zeroth-Order Optimization

For completeness, we provide the convergence of multi-perturbation ZO SGD to compare it with the learning dynamics point of view. To begin with, we need the following key lemma about the fourth-moment of Gaussian vector.

**Lemma 2** (Fourth-Order Moment of Gaussian Vector). *Suppose that the random vector $u \sim \mathcal{N}(0, I)$, i.e., drawing from a standard Gaussian distribution, for any symmetric matrix $W$,*

$$\mathbb{E}\, uu^\mathsf{T} W uu^\mathsf{T} = \mathrm{Tr}(W) \cdot I + 2W. \tag{24}$$

*Proof.* Let the matrix $\Psi = uu^\mathsf{T} W uu^\mathsf{T}$. For each element $i \neq j$,

$$\Psi[i, j] = \mathbb{E}\, u_i u_j (\sum_{i', j'} u_{i'} u_{j'} W[i', j']) = 2\mathbb{E}\, u_i^2 u_j^2 W[i, j] = 2W[i, j], \tag{25}$$

where the second equality holds because the zero-mean property of $z$ and $z_i$ is independent of each other. For the diagonal elements, we have

$$\begin{aligned}
\Psi[i, i] &= \mathbb{E}\, u_i^2 (\sum_{i', j;} u_{i'} u_{j'} W[i', j']) = \sum_{i'} \mathbb{E}\, u_i^2 u_{i'}^2 W[i', i'] \\
&= \sum_{i' \neq i} \mathbb{E}\, u_i^2 \mathbb{E}\, u_{i'}^2 W[i', i'] + \mathbb{E}\, u_i^4 W[i, i] \\
&= \sum_{i'} W[i', i'] + 2W[i, i] \cdot I,
\end{aligned} \tag{26}$$

where we utilize the fact that $\mathbb{E}\, z_i^4 = 3I^2$. Lastly, combining the above two results, we establish

$$\Psi = \mathrm{Tr}(W) \cdot I + 2W \tag{27}$$

□

Leveraging this lemma, we can derive the expectation for the multi-perturbation setting. Let the average projection matrix be denoted by $\bar{U} = \frac{1}{P} \sum_{p=1}^{P} u_{t,p} u_{t,p}^{\top}$. The expectation of the quadratic form expands as follows:

$$
\begin{aligned}
\mathbb{E}\left[\bar{U} W \bar{U}\right] &= \frac{1}{P^2} \sum_{p=1}^{P} \sum_{q=1}^{P} \mathbb{E}\left[u_{t,p} u_{t,p}^{\top} W u_{t,q} u_{t,q}^{\top}\right] \\
&\stackrel{(a)}{=} \frac{P^2 - P}{P^2} W + \frac{P}{P^2} \mathbb{E}\left[u_{t,p} u_{t,p}^{\top} W u_{t,p} u_{t,p}^{\top}\right] \\
&= \left(1 - \frac{1}{P}\right) W + \frac{1}{P}\left(\text{Tr}(W) \cdot I + 2W\right) \\
&= \left(1 + \frac{1}{P}\right) W + \frac{1}{P}\text{Tr}(W) \cdot I,
\end{aligned}
\tag{28}
$$

where we split the double summation into the case that $p = q$ and $p \neq q$ in the step (a).

In this paper, $u_{t,p} \in \mathbb{R}^d$ represents the perturbation vector, implying that $W \in \mathbb{R}^{d \times d}$ resides in a high-dimensional feature space. In the special case where $W = I$, the trace of the resulting expectation is $(1 + \frac{1+d}{P})d$, which scales quickly with the dimension $d$. As we will demonstrate shortly, this dimensional dependence is the primary source of the high variance observed in zeroth-order gradient estimates.

### C.1. The Convergence Rate under Standard Assumptions

Following standard conventions in the optimization literature, we adopt the following two assumptions regarding the loss landscape and the stochastic oracle.

**Assumption 1** ($L$-Smoothness). *The loss function $\ell(\theta)$ is $L$-smooth. That is, for all $\theta, \theta' \in \mathbb{R}^d$:*

$$
\ell(\theta') \leq \ell(\theta) + \langle \nabla \ell(\theta), \theta' - \theta \rangle + \frac{L}{2}\|\theta - \theta'\|^2.
\tag{29}
$$

□

**Assumption 2** (Unbiased Gradient with Bounded Variance). *We assume the algorithm accesses a stochastic oracle $\widehat{\ell}(\theta)$ whose gradient is an unbiased estimator of the true gradient with bounded variance. Specifically, for all $\theta$:*

$$
\mathbb{E}[\nabla\widehat{\ell}(\theta)] = \nabla\ell(\theta), \quad \text{and} \quad \mathbb{E}\|\nabla\widehat{\ell}(\theta) - \nabla\ell(\theta)\|^2 \leq \sigma^2.
\tag{30}
$$

□

The multi-perturbation ZO GD algorithm has the following form

$$
\begin{aligned}
\theta_{t+1} &= \theta_t - \eta \frac{1}{P} \sum_{p=1}^{P}\left[\frac{1}{2\mu}\left(\widehat{\ell}(\theta_t + \mu u_{t,p}) - \widehat{\ell}(\theta_t - \mu u_{t,p})\right) u_{t,p}\right] \tag{31} \\
&= \theta_t - \eta \frac{1}{P} \sum_{p=1}^{P} \langle \nabla\widehat{\ell}(\theta), u_{t,p}\rangle u_{t,p} + \mathcal{O}(\mu\eta) \tag{32} \\
&= \theta_t - \eta \frac{1}{P} \sum_{p=1}^{P} u_{t,p} u_{t,p}^{\mathsf{T}} \nabla\widehat{\ell}(\theta) + \mathcal{O}(\mu\eta) \tag{33} \\
&= \theta_t - \eta \left[\frac{1}{P} U_{t,P} U_{t,P}^{\mathsf{T}}\right] \nabla\widehat{\ell}(\theta) + \mathcal{O}(\mu\eta), \tag{34}
\end{aligned}
$$

where $\nabla\widehat{\ell}(\theta) \in \mathbb{R}^{d \times 1}$ is the stochastic gradient, and $U_{t,P} \in \mathbb{R}^{d \times P}$ is the horizontal stack of $P$ perturbation vectors. The second equality applies the following mean-value theorem and assumes the Hessian is bounded

$$
\widehat{\ell}(\theta_t + \mu u_{t,p}) - \widehat{\ell}(\theta_t - \mu u_{t,p})
$$

$$= \widehat{\ell}(\theta_t) + \mu\langle\nabla\widehat{\ell}(\theta), u_{t,p}\rangle + \frac{1}{2}\mu^2 u_{t,p}^\mathsf{T}\left(\int_0^1 \nabla^2\widehat{\ell}(\theta + au_{t,p})\mathrm{d}a\right)u_{t,p}$$

$$- \left(\widehat{\ell}(\theta_t) - \mu\langle\widehat{\nabla}\ell(\theta), u_{t,p}\rangle + \frac{1}{2}\mu^2 u_{t,p}^\mathsf{T}\left(\int_0^1 \nabla^2\widehat{\ell}(\theta - au_{t,p})\mathrm{d}a\right)u_{t,p}\right)$$

$$= 2\mu\langle\nabla\widehat{\ell}(\theta_t), u_{t,p}\rangle + \mathcal{O}(\mu^2) \tag{35}$$

Hence, the ZO gradient is an unbiased estimate of the true gradient (within that $\mathcal{O}(\mu^2)$ smoothness meaning). Without loss of generality, we assume that the entries of the perturbation vector $u_{t,p}$ are independent and identically distributed (i.i.d.) drawn from a standard Gaussian distribution. Under this assumption, it is straightforward to establish that:

$$\mathbb{E}\frac{1}{P}\sum_{p=1}^{P}u_{t,p}u_{t,p}^\mathsf{T} = \frac{1}{P}\sum_{p=1}^{P}\mathbb{E}\,u_{t,p}u_{t,p}^\mathsf{T} = \frac{1}{P}\sum_{p=1}^{P}I = I \tag{36}$$

Then, taking the conditional expectation on (34), we immediately obtain

$$\mathbb{E}_t\theta_{t+1} = \theta_t - \eta\nabla\ell(\eta_t) + \mathcal{O}(\mu\eta) \tag{37}$$

For the variance $\mathbb{E}_t\|\theta_{t+1} - \theta_t\|^2$, we will apply the lemma listed previously. Utilizing the Lipschitz condition, we have

$$\mathbb{E}_t\ell(\theta_{t+1}) \leq \ell(\theta_t) + \langle\nabla\ell(\theta_t), \mathbb{E}_t(\theta_{t+1} - \theta_t)\rangle + \frac{L}{2}\mathbb{E}_t\|\theta_{t+1} - \theta_t\|^2$$

$$= \ell(\theta_t) - \eta\|\nabla\ell(\theta_t)\|^2 + O(\eta^2\mu) + \frac{\eta^2 L}{2}\mathrm{Tr}\left\{\mathbb{E}_t\left[\left(\frac{1}{P}\sum_{p=1}^{P}u_{t,p}u_{t,p}^\mathsf{T}\right)\nabla\widehat{\ell}(\theta)\nabla\widehat{\ell}(\theta)^\mathsf{T}\left(\frac{1}{P}\sum_{p'=1}^{P}u_{t,p'}u_{t,p'}\right)\right]\right\}$$

$$\leq \ell(\theta_t) - \eta\|\nabla\ell(\theta_t)\|^2 + \underbrace{\frac{\eta^2 L}{2}(1 + \frac{d+1}{P})\|\nabla\widehat{\ell}(\theta_t)\|^2}_{\text{ZO Grad. Var. (Worst Case)}} + O(\eta^2\mu) \tag{38}$$

Assuming $d \gg P$, we have $1 + \frac{d+1}{P} \approx \frac{d}{P}$. Re-arranging terms and taking the expectation over stochastic noise, we obtain

$$\eta\left(1 - \frac{\eta Ld}{2P}\right)\|\nabla\ell(\theta_t)\|^2 \leq \ell(\theta_t) - \mathbb{E}_t\ell(\theta_{t+1}) + \frac{\eta^2 dL}{2P}\sigma^2 + O(\eta^2\mu) \tag{39}$$

If $\eta \leq \frac{P}{Ld}$, we establish the following sub-linear convergence rate

$$\frac{1}{T}\sum_{t=0}^{T-1}\mathbb{E}\|\nabla\ell(\theta_t)\|^2 \leq \frac{2}{\eta T}(\ell(\theta_0) - \ell^\star) + \frac{\eta dL}{P}\sigma^2 + O(\eta\mu) \tag{40}$$

For sufficiently large $T$ and let $\eta = \mathcal{O}(\sqrt{\frac{P}{dLT}})$, we have the convergence rate

$$\frac{1}{T}\sum_{t=0}^{T-1}\mathbb{E}\|\nabla\ell(\theta_t)\|^2 \leq \mathcal{O}\left(\sqrt{\frac{dL}{PT}}\right) \tag{41}$$

From this perspective, it is clear that under the fixed iterations, the larger perturbation $P$ leads to better performance. This is not a surprising conclusion. However, if we calculate the total query complexity, i.e., the total iterations multiplied by the perturbations per iteration, we have

$$\text{Query Complexity} = \mathcal{O}\left(\sqrt{\frac{dPL}{T}}\right) \tag{42}$$

### C.2. Low Effective Rank Assumption

Improvements to this bound are possible under the low-effective rank assumption, which can enhance the worst-case convergence rate. The primary challenge, however, is that the effective rank is computationally expensive to obtain.

Consequently, this assumption offers little practical guidance for selecting an appropriate ZO method.

The effective rank of Hessian is defined as $\text{Tr}(H)/\|H\|_2$, that is the sum of the eigenvalues divided by the maximum of the eigenvalues of $H$. The low-effective rank assumption of Hessian is

**Assumption 3** (Low Effective Rank). *There exists a Hessian matrix $H(\theta) \preceq L \cdot I_d$, such that $\nabla^2 \ell(\theta') \preceq H(\theta)$ for any $\theta'$ in a small neighborhood of $\theta$, and the effective rank of $H$ satisfies $\text{Tr}(H)/\|H\|_2 \leq \kappa$.* $\qquad\square$

Several indirect evidences indicate that $\kappa \ll d$ (Papyan, 2020; Yao et al., 2020; Malladi et al., 2023a). The looseness of the bound comes from the $L$-smoothness usage. Instead of using the bounds of Hessian, we can expand the iteration while keeping the Hessian. In this way, we can leverage the low effective rank assumption.

$$\mathbb{E}_t \ell(\theta_{t+1}) \leq \ell(\theta_t) + \langle \nabla\ell(\theta_t), \mathbb{E}_t(\theta_{t+1} - \theta_t)\rangle + \frac{1}{2}\mathbb{E}_t\left((\theta_{t+1} - \theta_t)^\mathsf{T} H_t(\theta_{t+1} - \theta_t)\right)$$

$$\leq \ell(\theta_t) - \eta\|\nabla\ell(\theta_t)\|^2 + \frac{\eta^2}{2}\left((1 + \frac{1}{P})\|\nabla\widehat{\ell}(\theta_t)\|_{H_t}^2 + \frac{1}{P}\text{Tr}(H_t)\|\widehat{\ell}(\theta_t)\|^2\right) + O(\eta^2\mu), \tag{43}$$

where $\|\nabla\widehat{\ell}(\theta_t)\|_{H_t}^2 = \nabla\widehat{\ell}(\theta_t)^\mathsf{T} H_t \nabla\widehat{\ell}(\theta_t)$. To leverage the low-effective rank assumption, we can treat this term through

$$\|\nabla\widehat{\ell}(\theta_t)\|_{H_t}^2 \leq L\text{Tr}(H/\|H\|_2\nabla\widehat{\ell}(\theta_t)\nabla\widehat{\ell}(\theta_t)^\mathsf{T}) \leq L\kappa\|\nabla\widehat{\ell}(\theta_t)\|^2.$$

Substituting back and similarly assuming the $\kappa \gg P$, we arrive

$$\eta\left(1 - \frac{\eta L\kappa}{2P}\right)\|\nabla\ell(\theta_t)\|^2 \leq \ell(\theta_t) - \mathbb{E}_t\ell(\theta_{t+1}) + \frac{\eta^2\kappa L}{2P}\sigma^2 + O(\eta^2\mu) \tag{44}$$

Now we have a bigger stability range for the learning rate $\eta \leq \frac{P}{L\kappa}$ to establish the following sub-linear convergence rate

$$\frac{1}{T}\sum_{t=0}^{T-1}\mathbb{E}\|\nabla\ell(\theta_t)\|^2 \leq \frac{2}{\eta T}(\ell(\theta_0) - \ell^\star) + \frac{\eta\kappa L}{P}\sigma^2 + O(\eta\mu) \tag{45}$$

For sufficiently large $T$ and let $\eta = \mathcal{O}(\sqrt{\frac{P}{dLT}})$, we finally get the convergence rate under the low-effective rank assumption:

$$\frac{1}{T}\sum_{t=0}^{T-1}\mathbb{E}\|\nabla\ell(\theta_t)\|^2 \leq \mathcal{O}\left(\sqrt{\frac{\kappa L}{PT}}\right) \tag{46}$$

Because $\kappa \ll d$, this bound typically is much better than the one in (41). If $\kappa$ is independent of $d$, above bound is also referred as dimension-free one.

## D. Experiment Setup and Extra Empirical Results

In this section, we provide more experiments setup details and empirical results to support our findings and conclusions in our main paper. To empirically validate our theoretical findings regarding the learning dynamics of ZO optimization, we conduct a series of experiments analyzing the approximation fidelity of the ZO eNTK. If the ZO eNTK can approch FO eNTK (i.g., low error, or high visual similarity), we will regard it as a good ZO approximation. We compute the exact FO eNTK as the ground truth baseline. We then compare this against the ZO eNTK estimated using stochastic perturbations. Our experiment setup are provided as follows:

### D.1. Group 1: LeNet Model, MNIST Dataset.

Our experiments start with some examples on the relatively low-dimensional LeNet ($d = 29,624$) model trained on the MNIST dataset. Unless otherwise stated, we employ standard Gaussian perturbations $u \sim \mathcal{N}(0, I)$. We systematically vary the number of perturbations $P$ (e.g., $P \in \{1, 5, 25, 125\}$) to observe the convergence trajectory of the ZO kernel towards the FO limit. To intuitively demonstrate the geometric alignment, we visualize the $V \times V$ kernel matrix (where $V = 10$ is the number of classes) as heatmaps, comparing the structural similarity between FO and ZO updates across different input pairs.

To rigorously quantify the approximation fidelity observed in the heatmaps, we track the convergence trajectory of the ZO eNTK as a function of the perturbation budget $P$. We utilize the relative Frobenius error to measure the distance between

the ZO eNTK approximation and the ground-truth FO eNTK. Formally, for a given pair $(\mathbf{x}_u, \mathbf{x}_o)$, this is defined as:

$$\text{Relative Frobenius Error} = \frac{\|\mathcal{K}_{ZO}(\mathbf{x}_u, \mathbf{x}_o; U_{t,P}) - \mathcal{K}_{FO}(\mathbf{x}_u, \mathbf{x}_o)\|_F}{\|\mathcal{K}_{FO}(\mathbf{x}_u, \mathbf{x}_o)\|_F}.$$

Moreover, in Figure 4, we utilize two extra metrics to measure the distance between ZO and FO eNTKs, including Centered Kernel Alignment (CKA) error and Spectral Distance. Specifically,

$$
\begin{aligned}
\text{CKA Error} =& 1 - \text{CKA}\big(\mathcal{K}_{FO}(\mathbf{x}_u, \mathbf{x}_o), \mathcal{K}_{ZO}(\mathbf{x}_u, \mathbf{x}_o; U_{t,P})\big) \\
=& 1 - \frac{\text{HSIC}\big(\mathcal{K}_{FO}(\mathbf{x}_u, \mathbf{x}_o), \mathcal{K}_{ZO}(\mathbf{x}_u, \mathbf{x}_o; U_{t,P})\big)}{\sqrt{\text{HSIC}\big(\mathcal{K}_{FO}(\mathbf{x}_u, \mathbf{x}_o), \mathcal{K}_{FO}(\mathbf{x}_u, \mathbf{x}_o)\big) \cdot \text{HSIC}\big(\mathcal{K}_{ZO}(\mathbf{x}_u, \mathbf{x}_o; U_{t,P}), \mathcal{K}_{ZO}(\mathbf{x}_u, \mathbf{x}_o; U_{t,P})\big)}},
\end{aligned}
$$

where HSIC is Hilbert-Schmidt Independence Criterion. We use this to measure the similarity between ZO-eNTK and FO-eNTK in terms of their ability to represent geometric structure.

To quantify the discrepancy between the eigenvalue distributions of the ZO and FO kernels, we employ the Spectral Distance, calculated as the 1-Wasserstein distance. Let $\boldsymbol{\lambda}^{(ZO)}$ and $\boldsymbol{\lambda}^{(FO)}$ denote the sets of eigenvalues for the kernel matrices $\mathcal{K}_{ZO}(\mathbf{x}_u, \mathbf{x}_o; U_{t,P})$ and $\mathcal{K}_{FO}(\mathbf{x}_u, \mathbf{x}_o)$, respectively, sorted in non-decreasing order (i.e., $\lambda_1 \le \lambda_2 \le \cdots \le \lambda_V$). The metric is defined as the mean absolute difference between the sorted spectra:

$$\text{Spectral Distance}\big(\mathcal{K}_{ZO}(\mathbf{x}_u, \mathbf{x}_o; U_{t,P}), \mathcal{K}_{FO}(\mathbf{x}_u, \mathbf{x}_o)\big) = \frac{1}{V} \sum_{i=1}^{V} \left| \lambda_i^{(ZO)} - \lambda_i^{(FO)} \right|$$

This metric assesses whether the ZO approximation preserves the global scaling and conditioning of the learning dynamics, which are inherently governed by the kernel's spectrum.

Unlike the discrete snapshots used for visualization, we evaluate the error across a continuous range of perturbation counts $P$, extending from 1 to $1,000$. This allows us to verify the asymptotic behavior of the error decay. We plot individual convergence curves for each observing sample $\mathbf{x}_o$ (classes $0 - 9$) against the fixed update sample $\mathbf{x}_u$. This stratification allows us to decouple the impact of sample similarity on convergence speed, distinguishing between "easy" (high similarity) and "hard" (low similarity) alignment tasks.

### D.2. Group 2: OPT Models, SST2 Dataset

To demonstrate the generality of our theoretical framework beyond low-dimensional networks, we extend our analysis to high-dimensional LLMs with varying parameter scales. We employ the OPT (Zhang et al., 2022) model family, specifically evaluating OPT-125M, 350M, and 1.3B. This progression allows us to empirically verify the dimension-free property of ZO learning dynamics, as the parameter count $d$ increases by an order of magnitude while the output vocabulary dimension $V$ remains constant. We conduct experiments on the SST-2 sentiment classification task (Socher et al., 2013; Wang et al., 2018). Critically, instead of replacing the final layer with a task-specific classification head, we formulate the task using a prompt-based approach (i.e., next-token prediction). This design choice ensures that the model's output dimension remains the full vocabulary size $V$ (e.g., $V \approx 50,272$ for OPT), rather than being reduced to the number of classes. This setup strictly aligns with our theoretical assumption where the output dimension $V$ is invariant to the model scaling. We compare the training trajectory of FO SGD against ZO SGD. For the ZO optimizer, we evaluate two perturbation distributions: Gaussian and Rademacher, to validate our theoretical claims regarding distribution efficiency. We vary the number of perturbations $P \in \{1, 5, 10, 20, 50, 100\}$ to observe how the ZO trajectory asymptotically approaches the FO baseline.

### D.3. Extra LLM Experiment Results.

We further validate our findings through additional LLM experiments on five distinct samples (shown in the yellow text box below). The corresponding model logit trajectories and output belief distributions are visualized in Figure 6 and Figure 7, respectively.

As illustrated in Figure 6, where the solid line is the ZO trajectory and the transparent line is the FO trajectory, the discrepancy between the ZO and FO probability distributions consistently decreases as the number of perturbations $P$ increases. This trend confirms that, given a sufficient perturbation budget, the ZO training trajectory effectively converges to that of standard FO methods. Moreover, Figure 7 also shows that the choice of perturbation distribution (Rademacher v.s. Gaussian) has a negligible impact on long-term convergence. While there may be slight variances at lower perturbation counts, both methods

converge to nearly identical $\ell_2$ as $P$ goes beyond 10, suggesting that ZO optimization is robust to any type of random noise used for gradient estimation. Furthermore, these two trends discussed above are remarkably consistent across varying model sizes. Although the likelihood region of OPT-1.3B model has a wider spread at lower perturbation numbers, the difference and spread decrease and become the same as smaller models as the number of perturbations increases to 100.

**Sample 1:** well-nigh unendurable ... though the picture strains to become cinematic poetry , it remains depressingly prosaic and dull .
**Sentiment: Negative**

**Sample 2:** a big , gorgeous , sprawling swashbuckler that delivers its diversions in grand , uncomplicated fashion .
**Sentiment: Positive**

**Sample 3:** it 's not that kung pow is n't funny some of the time – it just is n't any funnier than bad martial arts movies are all by themselves , without all oedekerk 's impish augmentation.
**Sentiment: Negative**

**Sample 4:** the jabs it employs are short , carefully placed and dead-center .
**Sentiment: Positive**

**Sample 5:** so unremittingly awful that labeling it a dog probably constitutes cruelty to canines .
**Sentiment: Negative**

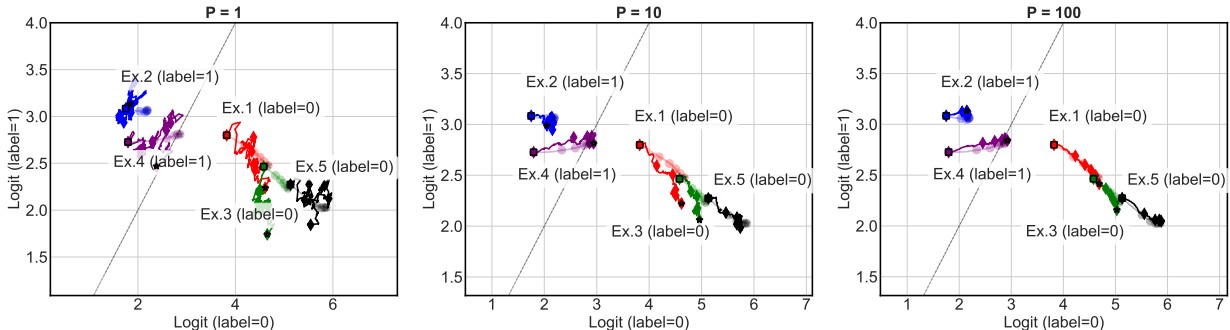

*Figure 6.* Logit Trajectories on Different $\mathbf{x}_o$ over 200 iterations.

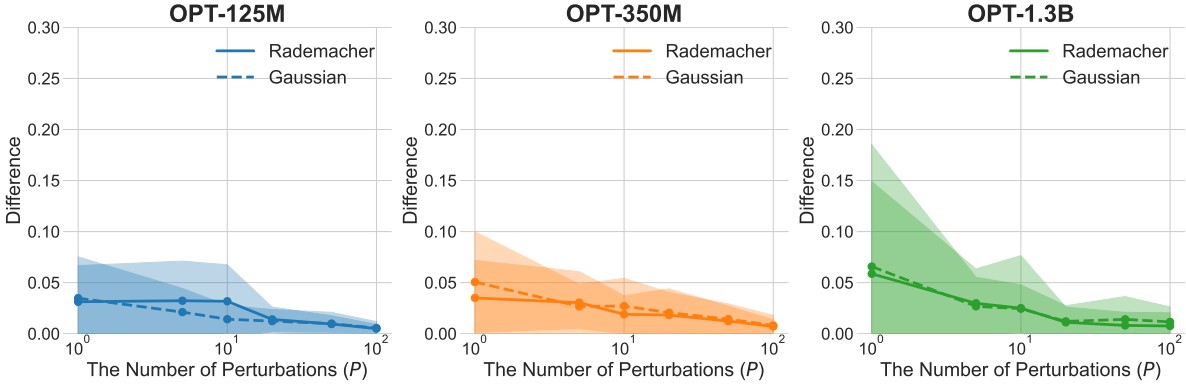

*Figure 7.* $\ell_2$ Norm Difference between ZO and FO Model Belief

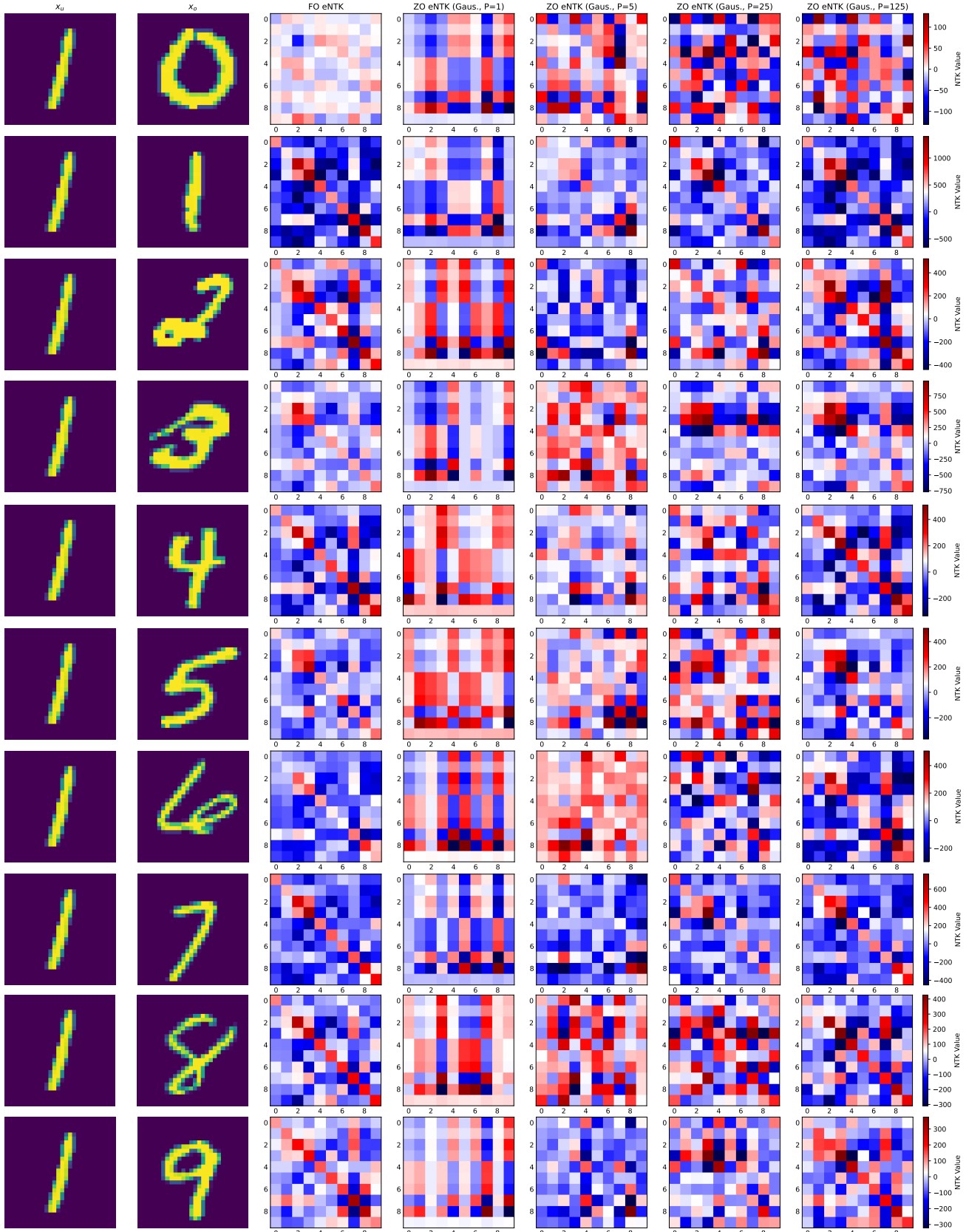

*Figure 8.* ZO eNTK v.s. FO eNTK under different test samples $\mathbf{x}_o$ and a fixed $\mathbf{x}_u = 1$. (LeNet, MNIST)

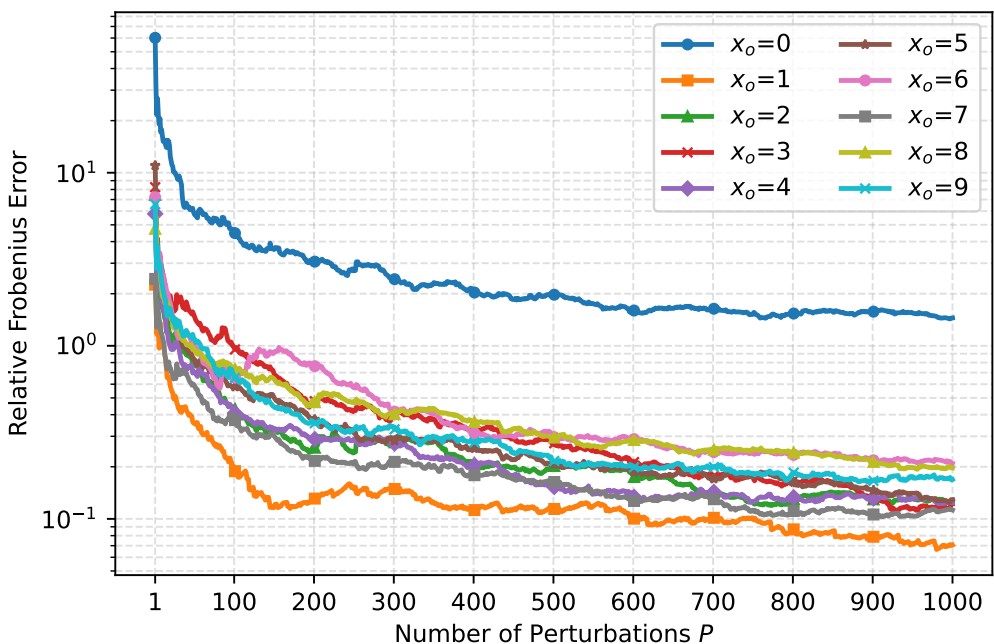

*Figure 9.* $\mathbf{x}_u = 1$ (LeNet, MNIST)

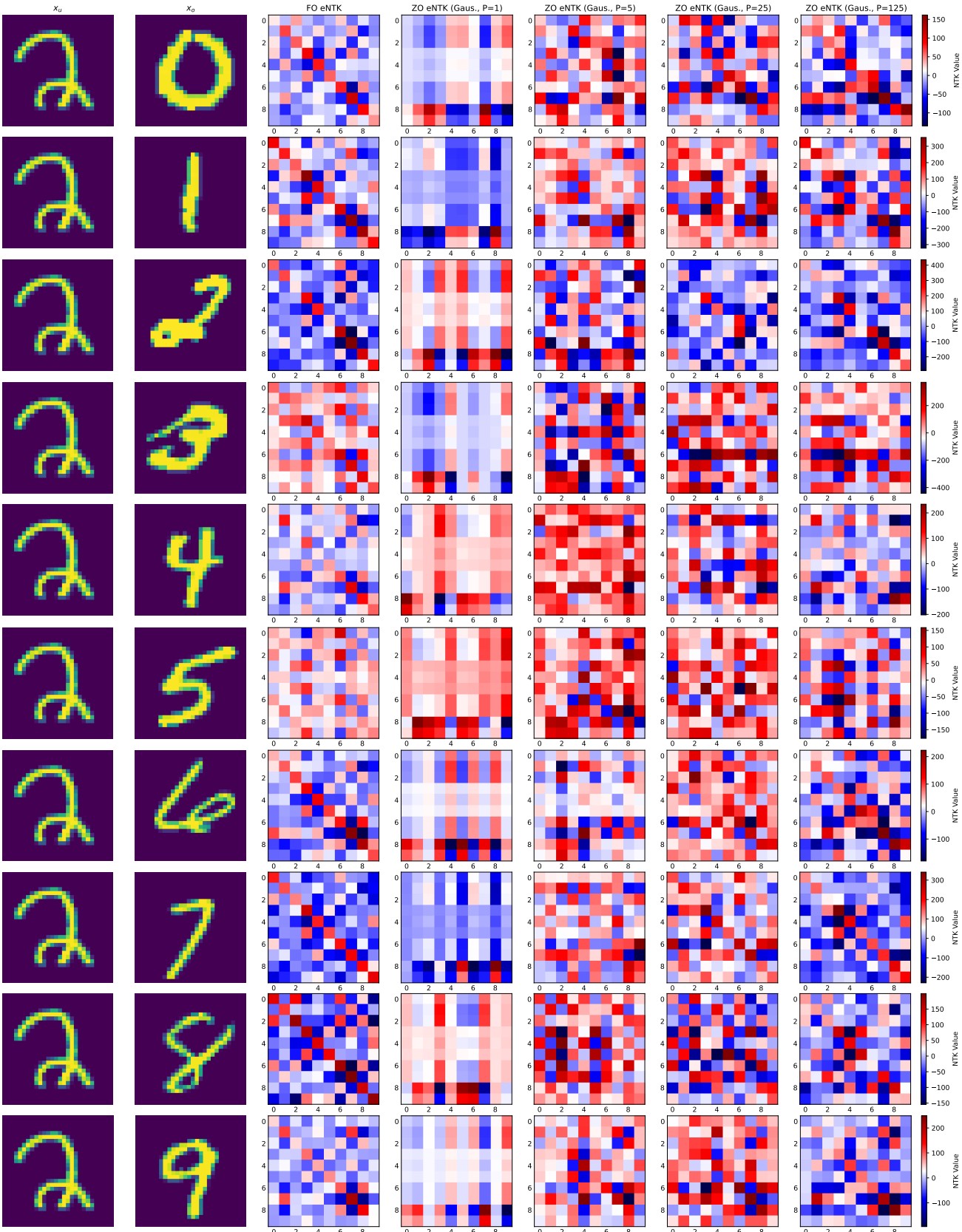

*Figure 10.* ZO eNTK v.s. FO eNTK under different test samples $\mathbf{x}_o$ and a fixed $\mathbf{x}_u = 2$. (LeNet, MNIST)

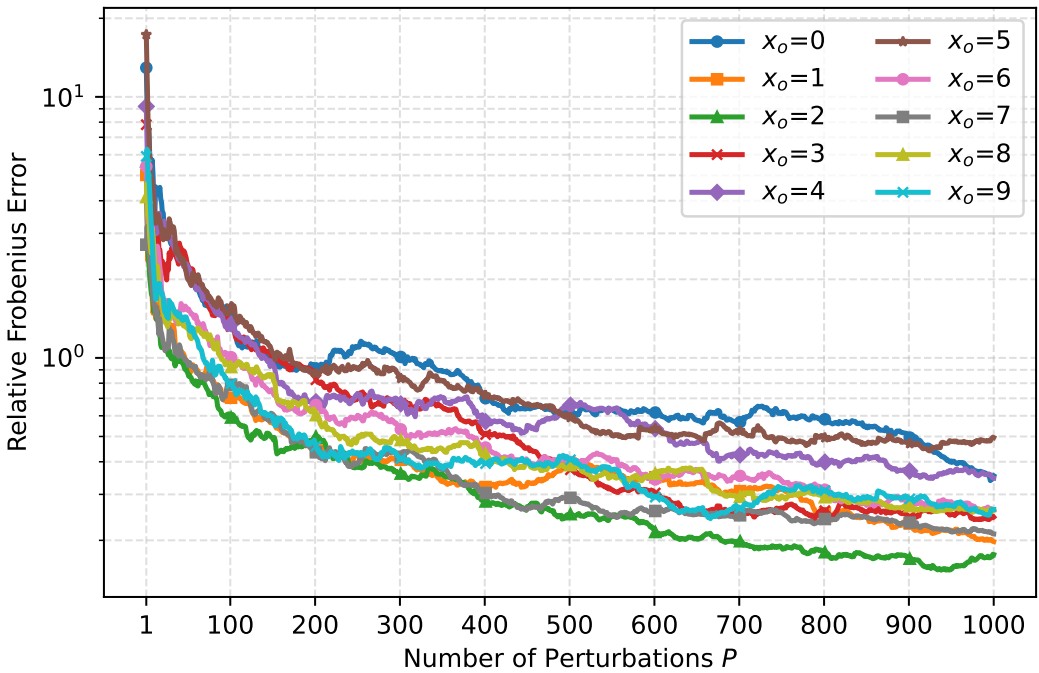

*Figure 11.* $\mathbf{x}_u = 2$ (LeNet, MNIST)

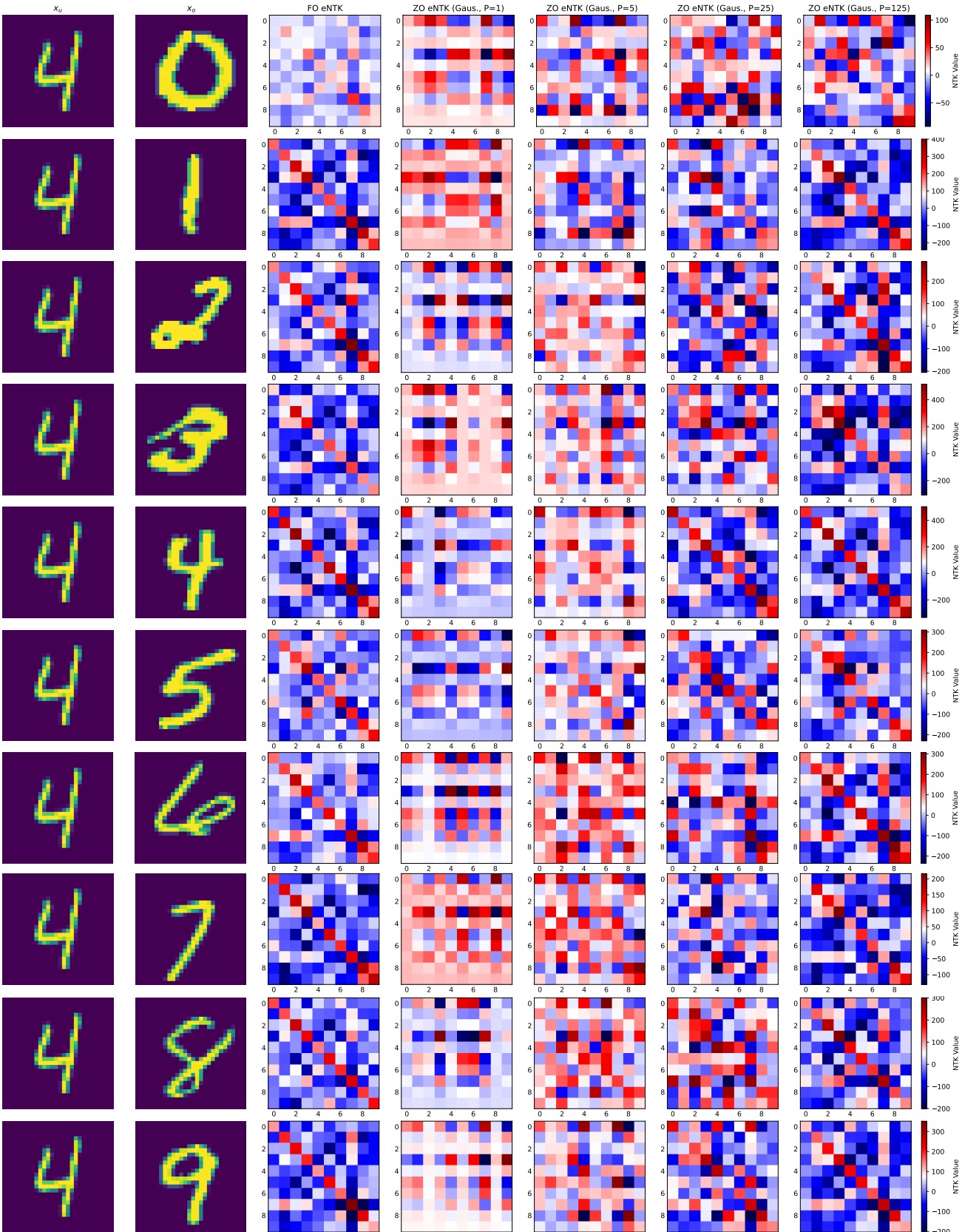

*Figure 12.* ZO eNTK v.s. FO eNTK under different test samples $\mathbf{x}_o$ and a fixed $\mathbf{x}_u = 4$. (LeNet, MNIST)

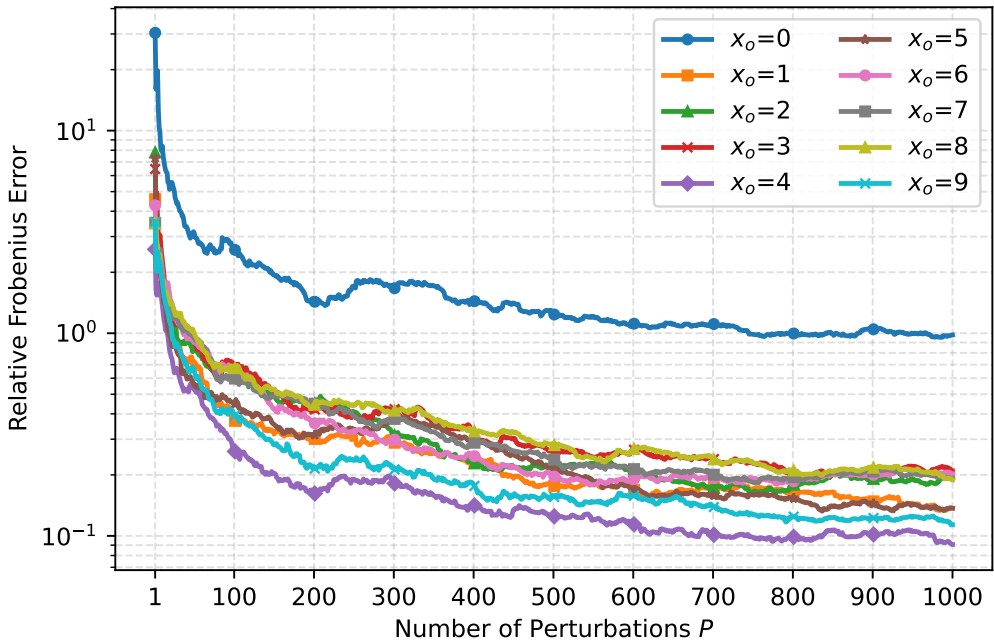

*Figure 13.* $\mathbf{x}_u = 4$ (LeNet, MNIST)

