# OpenReview forum: "Learning Dynamics of Zeroth-Order Optimization: A Kernel Perspective"
_ICML.cc/2026/Conference — ICML 2026 regular_

### Official Review · Reviewer_adav · 2026-02-13

**Soundness:** 3
**Presentation:** 4
**Significance:** 3
**Originality:** 4
**Overall Recommendation:** 5
**Confidence:** 3

**Summary:**

This work aims to address an important question in zeroth-order optimization; that is, the empirical success of these methods are not supported by existing theoretical analysis. The main paper contains two parts: One is from the optimization view; this perspective is standard but cannot explain the success of zeroth-order methods. Another part is the kernel perspective; it relies on the J-L lemma and results in a new conclusion w.r.t. the dependence of the dimension. Some empirical results are provided to validate these statements.

**Compliance With Llm Reviewing Policy:**

Affirmed.

**Final Justification:**

My concerns are fully addressed as indicated in my review. And I have raised my score from 4 to 5.

**Key Questions For Authors:**

1. To reflect the key finding of this work, the experiment should control the value of V instead of P. According to the eNTK point of view, as V decreases, the performance of ZO methods would be better and better; is my understanding correct? Can the author explain it further or empirically validate this statement?

2. Continue to the previous question, the output of a LLM seems to be large. The current theoretical analysis doesn't explain the success of the Zo methods in LLM. Some additional synthetic experiments might be helpful in validating the claim made in this paper.

3. As mentioned in the weaknesses point 3, the eNTK point of view seems to work for all LLMs and it is not limited to pre-trained LLMs. If I use P=100, is it possible to pre-train a LLM from a random initialization? Please correct me if I misunderstand something.

Overall, I am satisfied with this work due to its obvious originality and significance and I rated 4. If experiments are further improved and carefully designed to support the statement of this paper, I will raise the score to 5.

**Limitations:**

yes

**Strengths And Weaknesses:**

Strengths: This work addresses a critical problem in zeroth-order optimization that the dimension-free convergence of ZO methods in training LLMs. These results sound to me.

The presentation is clear. All important notations are labeled in different colors. The motivation is well supported.

Originality: The conclusion is very new, especially on the kernel perspective.

Weaknesses:

I believe this work has a solid result. But the empirical experiment doesn't support it well. It didn't tell how the output dimension V affects the peformance.

1. It has been known that the performance of LLM can be improved by scaling up P. So the statement " as P increases from 1 to 100, the ZO trajectories (for
both Gaussian and Rademacher distributions) progressively
align with the FO baseline. " is not something new.

2. The author claimed "This empirical evidence strongly
corroborates our kernel-based derivation: the fidelity of ZO
learning dynamics is governed by the output dimension V
(which remains constant across these models), rather than
being diluted by the massive expansion of d." on page 8. I don't understand how this figure supports this statement. See my questions below.

3. There is still a gap between the kernel point of view and the existing empirical observations. "Malladi et al. (2023b) proposed the low-effective rank assumption" and they claim that the pre-trained LLM will satisfy this assumption. However, the eNTK point of view seems to work for all neural networks; it doesn't tell the difference between pre-trained and randomized LLMs.

---

> ### Author Rebuttal · Authors · 2026-03-30
>
> We sincerely appreciate the reviewer's thorough review and inspiring questions. The following is our responses.
>
> > W1: improved by scaling up P is not something new
>
> We agree that the empirical benefit of scaling $P$ is well-established. However, our novelty lies in explaining why this scaling works for billion-parameter LLMs by analyzing the problem through **a fundamentally different lens**: learning dynamics in function space. Classical optimization theory views scaling $P$ as a way to reduce gradient variance in parameter space. Yet, these traditional bounds heavily depend on the model dimension $d$ (e.g., $\mathcal{O}(d/P)$). This creates a "dimension paradox"—if the convergence bound scales linearly with $d$, ZO methods should theoretically fail entirely for massive LLMs. By analyzing the eNKT, we show that increasing $P$ acts as a geometric projection that reconstructs the FO trajectory. The core novelty here is our mathematical proof (Equation 17) establishing that the required $P$ for trajectory alignment depends solely on the output dimension $V$, completely decoupled from the massive parameter dimension $d$. **This shift from parameter space to function space** resolves the dimension paradox, providing a new, dimension-free theoretical justification for ZO's scalability.
>
> > W2 & Q1 & Q2 & Q3: Performance and experiment over V and Practical Success in Fine-tuning.
>
> We sincerely thank the reviewer for these deeply connected and constructive questions regarding the role of the output dimension $V$. Your intuition is exactly correct: our theory predicts that a smaller $V$ improves the fidelity of the ZO-eNTK, leading to better optimization performance. It is a little bit tricky to give apples-to-apples validation over different $V$ since different $V$ typically indicate different models or different tasks. The followings are our best efforts to address the question
>
> 1. **Theoretical Explanation & LLM Practicality** In our framework (Equation 17), the approximation error of the ZO gradient estimator is fundamentally governed by the relationship between the output dimension $V$ and the number of perturbations $P$. A smaller $V$ reduces the complexity of the output manifold, meaning fewer random perturbations are required to capture the "informative" directions of the gradient with high fidelity. How does this explain ZO success in LLMs? While the native vocabulary of an LLM is indeed massive (e.g., ~50,000), practical ZO fine-tuning typically utilizes prompt-based tuning with verbalizers. This approach creates a representational bottleneck that drastically reduces the effective $V$ from the entire vocabulary down to a minimal set of task-specific labels (e.g., $V=2$ for binary classification). This is precisely what enables ZO methods to scale: they succeed in regimes where the parameter dimension $d$ is massive, but the effective output dimension $V$ is kept small.
>
> 2. **Synthetic Data Validation:** To explicitly validate our theoretical bound—specifically that ZO fidelity degrades at a rate of $\mathcal{O}(\sqrt{V \log V / P})$—without the confounding architectural variables of an LLM, we set up a controlled synthetic experiment. We constructed a two-layer MLP with a fixed hidden dimension and a fixed number of perturbations ($P=50$). We scaled $V$ from 2 to 1000 and measured the Relative Frobenius Error between FO-eNTK and ZO-eNTK.
> As shown in Table 2, the approximation error increases at a rate consistent with $\sqrt{V}$, exactly as predicted by our theorem.
>
> Table 2: ZO eNTK Approximation Error vs. Output Dimension ($V$)
>
> | Output Dimension (V) | Difference Norm (k) | FO Norm (f) | Relative Error |
> |----------------------|---------------------|-------------|----------------|
> | 2                    | 21.2548             | 76.8934     | 0.2764         |
> | 10                   | 81.5816             | 184.9210    | 0.4412         |
> | 100                  | 957.8461            | 653.4963    | 1.4657         |
> | 500                  | 5083.9067           | 1706.7668   | 2.9787         |
> | 1000                 | 6830.7114           | 1849.7047   | 3.6929         |
>
> 3. **Empirical Validation on Yahoo! Answers** We further conducted a new experiment using the Yahoo! Answers Topic dataset (a 10-class text classification task). Using an OPT-350M model with a fixed $P=20$, we controlled the effective $V$ by restricting the fine-tuning task to the first 2, 5, and all 10 classes.As shown in Table 1, the ZO method's performance monotonically improves as $V$ shrinks, perfectly aligning with your hypothesis.
>
> Table 1: Per class accuracy at iteration 200. P=20, Model=OPT-350M.
> | Dataset  | Class 0  | Class 1  | Class 2  | Class 3  | Class 4  |
> | -------- | -------- | -------- | -------- | -------- | -------- |
> | Yahoo-2  | 82%      |  85%     | N/A      | N/A      | N/A      |
> | Yahoo-5  | 58%      |  69%     | 80%      | 32%      | 80%      |
> | Yahoo-10 | 26%      |  66%     | 76%      | 31%      | 54%      |

---

> > ### Author Rebuttal · Reviewer_adav · 2026-04-01
> >
> > My concerns are fully resolved and I will raise my score to 5.

---

> > > ### Author Response · Authors · 2026-04-05
> > >
> > > Thank you so much for your positive feedback. We are pleased that we successfully addressed your concerns and questions. If your have other questions, we are happy to provide further explanation.

---

### Official Review · Reviewer_SPRN · 2026-03-09

**Soundness:** 3
**Presentation:** 3
**Significance:** 3
**Originality:** 3
**Overall Recommendation:** 4
**Confidence:** 3

**Summary:**

This paper investigates the learning dynamics of zeroth-order optimization using
empirical neural tangent kernel (NTK) perspective.
They show that zeroth-order optimization can be viewed as an approximation of the first-order optimization with an approximated NTK, which is a random projection of the original NTK.
By considering the learning dynamics in the function space, convergence rates independent of the number of parameters are obtained.
Theoretical results are supported by experiments on toy examples.

**Compliance With Llm Reviewing Policy:**

Affirmed.

**Final Justification:**

My concerns have been resolved. Thus, I will maintain my positive score.

**Key Questions For Authors:**

1. Is it possible to obtain convergence rates independent of the number of outputs as well under some additional assumptions? For example, by assuming some structural properties of the NTK matrix?
2. Instead of sampling from isotropic distribution, is it possible to sample from a distribution that is more aligned with the NTK matrix to obtain better convergence rates?

**Limitations:**

yes

**Strengths And Weaknesses:**

**Strengths:**
- The paper is well-written and well-structured. The authors clearly explain the motivation behind their work and the main contributions.
- This paper explains the practical success of zeroth-order optimization to some extent
by showing ZO can avoid the curse of dimensionality in terms of the number of parameters.
- Kernel perspective on learning dynamics of zeroth-order optimization is interesting and can potentially be applied to other optimization algorithms as well.

Note: I am not confident about the novelty of this paper as I am not familiar with the literature on zeroth-order optimization.

**Weaknesses:**
- Theoretical analysis is based on the first-order approximation of the optimization dynamics, which may not be accurate in practice.
- Models considered in the experiments are relatively small (up to 1.3B).
- While the number of outputs is relatively small compared to the number of parameters, it is still large and thus it does not fully explain the practical success of zeroth-order optimization in large-scale models.

---

> ### Author Rebuttal · Authors · 2026-03-30
>
> We sincerely thank the reviewer for the insightful points.
>
> ---
> > W1: Inaccurate first-order approximation in practice.
>
> **Our response:** While linearizing optimization can be inaccurate when training from scratch, our analysis targets the fine-tuning regime, where this approximation is well justified. We support this with two points:
> 1. In fine-tuning LLMs, updates are very small since pre-trained weights are already near a good solution. The loss landscape is thus locally linear and well captured by a first-order Taylor expansion, an assumption standard in NTK-based analyses.
> 2. If this approximation were inaccurate, our projected kernel predictions would deviate from practice. Yet, Figure 5 shows that ZO-SGD closely tracks FO-SGD across model scales (125M to 1.3B), confirming that the first-order perspective captures the basic optimization dynamics.
>
> ---
> > W2: Models scale.
>
> **Our response:** We conducted extra experiments on 2.7B LLMs, matching the scale in [1]. The results show that the trajectory alignment phenomenon persists: ZO-SGD continues to closely track FO-SGD, consistent with Figure 5. This suggests the dimension-independent convergence we observe isn't limited to smaller models, but also holds at larger scales. Due to rebuttal constraints, we are unable to include the new figures here, but we'll inlcude it in our future version.
>
> [1] Ren, Y. and Sutherland, D. J. Learning dynamics of LLM finetuning. In ICLR, 2025.
>
> ---
> > W3: On the scale of output dimension $V$.
>
> **Our response:** Traditionally, ZO methods, which estimate gradients using only loss values, were considered impractical for large-scale models since their theoretical convergence was tied to $d$ (the dimension curse), predicting that tuning a billion-parameter model would be millions of times slower than standard methods. Our work resolves this paradox by proving, through an eNTK lens, that ZO convergence is actually decoupled from $d$ and is instead governed by the output dimension $V$. Crucially, this theoretical dimension-independence is rigorously supported by our experiments, which scale across several orders of magnitude, from small-scale LeNet models to 1.3B LLMs. Across this entire range, we observe the same trajectory alignment phenomenon where ZO optimization closely tracks the first-order baseline, offering strong empirical evidence that our theory correctly identifies the underlying mechanism of ZO's success in practice. By shifting the bottleneck from the prohibitive $d$ to the manageable $V$, our work transforms ZO from a theoretically inefficient heuristic into a mathematically justified, scalable solution for the LLM era.
>
> ---
> > Q1: Convergence rates independent of $V$
>
> **Our response:** While our primary focus is resolving the dimension paradox by decoupling ZO convergence from the model dimension $d$, extending this to the output dimension $V$ is an intriguing direction. Although we do not yet have a formal theorem, we believe this is highly plausible and offer some supporting insights. From a practical perspective, in instruction fine-tuning tasks, the effective output space is much smaller than the full vocabulary. Although models like GPT-2 have $V \approx$ 50000, the task often only involves a few tokens (e.g., “positive” or “negative”). We hypothesize that this reduced effective $V$ is a key reason why ZO methods perform so well in these settings.
>
> On the theoretical side, the dependence on V arises from Eq. (15), where we bound $\\|\Delta\mathcal{K}_t \mathcal{G}_t\\|_2 \leq \\|\Delta\mathcal{K}_t\\|_F \\|\mathcal{G}_t\\|_2,$ without exploiting structure in the kernel discrepancy $\Delta \mathcal{K}_t$. A tighter analysis is possible. Noting $\\|\Delta\mathcal{K}_t \mathcal{G}_t\\|_2^2 = \mathcal{G}_t^\top \Delta\mathcal{K}_t^2 \mathcal{G}_t,$ this term reflects eigenvalue-weighted contributions. Since $\Delta\mathcal{K}_t^2$ is PSD, it can be bounded via its spectral norm, $\\|\Delta\mathcal{K}_t \mathcal{G}_t\\|_2^2 \leq \max \lambda(\Delta\mathcal{K}_t^2)\\|\mathcal{G}_t\\|_2^2,$ or tightened further using full eigenvalue expansion. Under structural assumptions (e.g., a rapidly decaying eNTK spectrum), one could potentially derive bounds independent of $V$. Another source of $V$-dependence comes from the JL lemma, yielding $P = \mathcal{O}(\log |V|),$ which is already mild and may be further improved via non-uniform sampling. Overall, removing the dependence on $V$ appears plausible and is a promising direction for future work.
>
> ---
> > Q2: On NTK-aligned sampling
>
> **Our response:** Theoretically, sampling aligned with the principal components of the NTK could improve gradient fidelity and speed up convergence. While promising, such geometry-aware sampling is beyond the scope of this work. Our goal is primarily descriptive: to explain why simple, widely-used strategies (e.g., Gaussian perturbations) are already effective for LLM fine-tuning. We will highlight informed sampling as a future direction.

---

> > ### Author Rebuttal · Reviewer_SPRN · 2026-04-01
> >
> > Thanks for the clarification. My concerns have been resolved.
> > Although this paper has several limitations (e.g., dependence on $V$ and the use of linear approximation), it provides an interesting new perspective on the analysis of zeroth-order optimization. Thus, I will maintain my positive score.

---

> > > ### Author Response · Authors · 2026-04-05
> > >
> > > We sincerely appreciate your positive feedback for our work and are pleased to know that we successfully addressed your concerns. If your have other questions, we are happy to provide further explanation.

---

### Official Review · Reviewer_99jG · 2026-03-11

**Soundness:** 3
**Presentation:** 3
**Significance:** 3
**Originality:** 3
**Overall Recommendation:** 4
**Confidence:** 4

**Summary:**

This paper studies when zeroth-order optimization can behave similarly to first-order training in fine-tuning settings. The main idea is to analyze how a ZO step changes model outputs, rather than only how well it estimates a gradient in parameter space. The authors show that the difference between ZO and FO updates can be written through a projected version of the model’s empirical kernel, and use this to argue that the quality of the approximation is driven mainly by the number of perturbations and the output dimension. The paper supports this with theory, small-scale kernel visualizations on LeNet/MNIST, and FO vs. ZO trajectory comparisons on OPT models for SST-2. Overall, the paper offers a new explanatory view of why multi-perturbation ZO can track FO behavior reasonably well in fine-tuning regime

**Compliance With Llm Reviewing Policy:**

Affirmed.

**Key Questions For Authors:**

- What is the strongest claim the authors believe their current theory supports beyond one-step update fidelity? The main result seems local, so I would appreciate a sharper statement of what should and should not be concluded about full training behavior.

- The limitations section says the current framework does not explicitly handle sequential reasoning settings. I think this is important, so can the authors state more directly in the main paper which task domains they believe are covered by the current theory and which are not?

- The LeNet/MNIST experiments are useful for illustrating the kernel story, but they also reveal substantial heterogeneity across pairs. How should we connect that observation to the much larger-model setting, where the same heterogeneity is harder to inspect directly?

**Limitations:**

yes

**Strengths And Weaknesses:**

Soundness:

- I think the empirical validation is better aligned with the theory than in many papers of this type. The LeNet/MNIST experiments are not presented as performance benchmarks, but as direct visualizations of the kernel object the theory studies. Then the OPT/SST-2 experiments move to trajectory comparisons on actual LLM fine-tuning. That two-level validation makes the paper more credible than if it had only shown downstream accuracy.
- The paper is careful to place its contribution next to the classical optimization view instead of pretending the old theory was wrong. It explicitly derives the standard worst-case ZO convergence rate with its d-dependence and contrasts it with the learning-dynamics bound that depends on V and P.
- The paper is very explicit about its own limits. It does not bury them in vague future work language. It directly names query-cost growth, memory problems in parallelization, failure to cover sequential reasoning, and likely mismatch to pre-training from scratch dynamics.

- The “dimension-free” message is narrower than the headline makes it sound. The dependence on parameter dimension d is replaced by a dependence on output size V, but V remains explicit in the bound. So this is not dimension-free in the literal sense. It is a change in which dimension matters
- The LLM validation is designed to match the theory by using prompt-based next-token prediction so that V stays fixed, which is fair, but also means the empirical setting is favorable to the theory being tested.


Presentation:

- The paper has a clear structure: it starts from one-step learning dynamics, introduces the projected-kernel view, then studies distribution choice, perturbation count, and finally LLM validation.
- The manuscript generally explains why it wants to study output-side learning behavior rather than only scalar loss.

- The paper uses “trajectory” language quite broadly, even though the formal control is one-step and the multi-step evidence is empirical.
- The rhetorical framing is broader than the formal scope in several places, especially where the paper talks about “resolving” the paradox.

Significance:

- An important concept analyzed by the paper is whether preserving the right prediction-space geometry is enough for ZO to follow FO learning dynamics.
- The contribution is potentially useful to both theory and practice because it gives a more structured explanation for why perturbation count matters.
- The significance is mostly conceptual, but it is still real. By moving the discussion from raw gradient recovery to preservation of output-side learning behavior, the paper gives a more usable way to interpret why some ZO fine-tuning regimes work at all.

- The main knob that improves the approximation (P), also increases function-query cost linearly.
- The paper itself says the framework does not naturally cover sequential reasoning tasks, which are central for many LLM uses.


Originality:

- The paper does not just benchmark ZO on LLMs again. It changes the analysis target.
- The side-by-side contrast between the optimization view and the learning-dynamics view is itself a useful conceptual contribution.
- The work combines theory, small-kernel visualization, and large-model trajectory experiments in service of one coherent interpretive claim

- The originality is strongest as a framework and interpretation, not as a fundamentally new mathematical tool.

---

> ### Author Rebuttal · Authors · 2026-03-30
>
> We sincerely thank the reviewer for the insightful and helpful comments and would like to reply to them as follows.
>
> ---
> > **Q1**: Strongest claim and what should and should not be concluded about full training behavior.
>
> **Our response:** The core contribution of our work is the discovery that the curse of dimensionality in zeroth-order (ZO) optimization is largely an artifact of the parameter-space perspective and does not fundamentally constrain function-space learning. Our new kernel-based analysis proves that in lazy learning regimes (e.g., LLM fine-tuning), the learning trajectory is governed by the output dimension $V$ (e.g., vocabulary size) rather than the massive model dimension $d$. We demonstrate that ZO optimization preserves the geometric alignment of the model's evolution in function space, effectively closing the long-standing gap between pessimistic classical theory and the empirical success of methods like MeZo.
>
> Regarding "what should not be concluded," we explicitly address these boundaries in Appendix B (Trade-Offs, Limitations, and Future Work):
> 1. Pre-training Dynamics: Our results should not be generalized to training from scratch. In such regimes, the eNTK is unstable, and random subspace projections may fail to track the rapidly evolving features during the early phase of feature learning.
> 2. Sequential Dependencies: Our analysis assumes the standard independent update structure. In tasks with complex dependencies, such as sequential reasoning (Chain-of-Thought), the error accumulation from ZO approximations may propagate non-linearly, as the classical independence assumptions regarding data may no longer hold.
>
> ---
> > Q2: On task domains and sequential reasoning.
>
> **Our response:** We study the standard empirical risk minimization (ERM) formulation (Sec. 2), and our results apply to tasks that fall within this setting. In particular, our theory is for supervised fine-tuning (SFT) tasks in the lazy training regim, where features remain approximately stable. It does not cover (i) tasks requiring training from scratch or substantial feature learning, and (ii) tasks with strong sequential dependencies, such as long-horizon reasoning. While these distinctions are discussed in Appendix B, we agree that stating them clearly in the main paper would improve clarity, and we will incorporate this in the revision.
>
> For clarity, we summarize the scope with concrete examples below.
> 1. **Covered domains**. Our framework captures standard SFT tasks such as classification, sentiment analysis, and short-form generation (e.g., prompt-based tuning with verbalizers). In these settings, the empirical NTK remains stable, and the effective output dimension ($V$) is small, enabling ZO-SGD to closely match FO-SGD with a limited number of perturbations.
> 2. **Non-covered domains**.
> (1) Sequential reasoning tasks: Long-form chain-of-thought or multi-step reasoning involves strong autoregressive dependencies across tokens, leading to compounding distribution shifts that are not captured by our current one-step kernel analysis. In addition, the effective output dimension grows with sequence length, which may require more perturbations or a different analysis.
> (2) Training-from-scratch settings: When models are trained from initialization, feature learning dominates and the kernel evolves significantly, violating the lazy training assumption underlying our theory.
>
> ---
> > Q3: Connecting kernel heterogeneity to large-scale models.
>
> **Our response:** The heterogeneity observed in the LeNet/MNIST experiments reflects the non-uniform geometry of the learned feature space (e.g., how different data pairs or classes interact). While direct inspection on LLMs is not possible, we can connect these small-scale observations to LLMs through the following two perspectives:
>
> 1. In large models, while we cannot compute or inspect the full $N \times N$ eNTK entry-wise, the optimization trajectory (Figure 5) serves as a reliable "aggregate proxy." If the underlying heterogeneity in LLMs were to negatively impact the projection fidelity, we would see a significant divergence between ZO-SGD and FO-SGD. The fact that ZO-SGD closely tracks FO-SGD across scales (125M to 1.3B) suggests that the aggregate kernel dynamics remain stable and the projection preserves the informative directions of the kernel.
> 2. In the context of the JL Lemma, heterogeneity often implies that the kernel matrix is highly structured or effectively low-rank. In pre-trained LLMs, this structure is likely even more pronounced than in LeNet due to the strong inductive biases learned during pre-training. Our theory shows that as long as the output dimension $V$ is manageable, the random perturbations $P$ will capture the dominant energy of this structured kernel. Thus, the heterogeneity in LeNet is likely the same structural property that enables success in LLMs.

---

> > ### Author Rebuttal · Reviewer_99jG · 2026-04-02
> >
> > Thank you for the clarifications. The rebuttal helps by stating the intended scope more explicitly, especially that the theory is aimed at supervised fine-tuning in a lazy-training regime and does not cover pre-training from scratch or settings with strong sequential dependencies. I appreciate that, and I agree these boundaries should be stated more clearly in the main paper.

---

> > > ### Author Response · Authors · 2026-04-05
> > >
> > > Thank you so much for your positive feedback for our paper, and we are glad to know that we successfully solved your concerns. If your have other questions, we are happy to provide further explanation.

---

### Official Review · Reviewer_wBCZ · 2026-03-12

**Soundness:** 2
**Presentation:** 3
**Significance:** 3
**Originality:** 3
**Overall Recommendation:** 3
**Confidence:** 4

**Summary:**

This paper studies why the two-point zeroth-order gradient estimator (ZO) works in practical LLM finetuning without suffering from a theoretical dimension-dependent convergence speed. It explains this gap by analyzing the learning dynamics of ZO via empirical Neural Tangent Kernel (eNTK), concluding that the approximation error is governed by the number of perturbations and model output size rather than the number of model parameters.

**Compliance With Llm Reviewing Policy:**

Affirmed.

**Final Justification:**

The rebuttal addresses W1 and W2. My main concern is the unresolved W3: I think it is not sound to use linear approximation in analyses. This paper's main contribution is theoretical analyses, and it claims "ZO methods can achieve a dimension-independent convergence rate even without the strict low-effective-rank assumption on the Hessian", which is essentially comparing to analyses in prior work such as [1,2]. However, these works do not use linear approximation. Instead, [2], for example, explicitly discusses the
$\mu$ value to take to obtain $d$-independent convergence rate. To make this paper's claim hold under a fair comparison to other work, I think the assumption of linear approximation should not be made.

[1] Fine-Tuning Language Models with Just Forward Passes, NeurIPS 2023.
[2] DPZero: Private Fine-Tuning of Language Models without Backpropagation. ICML 2024.

Therefore, I retain my original score.

**Key Questions For Authors:**

Please refer to the "Strengths and Weaknesses" section for my questions. If my questions are addressed, I would love to raise my score.

**Limitations:**

Yes, this paper has discussed the future work directions in the appendix.

**Strengths And Weaknesses:**

**Strengths**

Why ZO methods work on LLM finetuning tasks (with prompts in particular) has been an open question, and this paper does provide a novel perspective on this question. It leverages several theoretical tools to illustrate the effectiveness of increasing the number of perturbations from a new angle (the old angle is reducing ZO gradient variance).

However, I have several questions as follows.

**Weaknesses**

1. This paper claims “ZO methods can achieve a dimension-independent convergence rate even without the strict low-effective-rank assumption on the Hessian” in lines 413-415, which is a strong claim. It relaxes assumptions in prior ZO theoretical analyses, but I’m worried that those assumptions are needed to make the analyses consistent with empirical observations. For example, this paper uses $V$ to represent the vocabulary size or the number of classes, and error increases as $V$ increases (eq. 17). Then, is it contradictory with the fact that the performance of ZO finetuning with prompts is much better than finetuning without prompts? In fact, the two-point ZO gradient estimator is found not to work on LM finetuning without prompts (i.e., train $k$-class classification head, $k\ll V$, as suggested in [1]).
2. It is also observed that LM finetuning can work with very small $P$, i.e., number of perturbations each iteration [1], but vision models/tasks require large $P$ [2]. With relaxed assumptions, can the proposed kernel view explain this?
3. This paper’s analysis directly assumes that the first-order Taylor expansion is a good approximation of the function value, abandoning the smoothing radius $\mu$ in the analysis. I’m concerned because in both practice and theory, the convergence speed depends on how large $\mu$ is chosen. Large $\mu$ and $\eta$ speeds up convergence. Could the authors discuss why ignoring $\mu$ makes sense and what consequences it would have on their results if they were not ignored?
4. Could the authors specify whether and how the learning rate $\eta$ and the smoothing radius $\mu$ are tuned for the results in Figure 5? It is surprising to see that SGD is neither faster nor better than ZO even when $P=1$. Could the authors additionally try some more challenging GLUE datasets (such as SST-5 or NLI datasets) and see whether ZO still outperforms when $P=1$? In my own experience, the SST-2 dataset is simple, and the gap between methods is usually small and not very informative.

[1] Fine-Tuning Language Models with Just Forward Passes, NeurIPS 2023.
[2] DeepZero: Scaling up Zeroth-Order Optimization for Deep Model Training, ICLR 2024.

---

> ### Author Rebuttal · Authors · 2026-03-30
>
> We sincerely thank the reviewer for the insightful comments.
>
> > W1: Dependency on $V$ and prompt-based tuning
>
> **Our response:** We clarify that our result in Eq. 17 isn't contradictory to empirical observations; rather, it offers a rigorous theoretical explanation for why the success of ZO fine-tuning is so tightly coupled with the use of prompts. The core of our theory is that the gradient fidelity relies on the output dimension $V$, which varies depending on the tuning strategy:
>
> - Prompt-based tuning (small $V$): In this setting, a verbalizer restricts the model's output to a few task-specific labels (e.g., "good" or "bad"). This effectively collapses the output dimension from the full vocabulary (e.g., 50,000) to a tiny set (e.g., $V=2$). According to Eq. 17, this reduction in $V$ drastically increases the fidelity of the ZO estimator, explaining why prompt-based ZO is so effective.
>
> - Tuning without prompts (large $V$): In naive fine-tuning (e.g., training a classification head over the full vocabulary or next-token prediction), $V$ remains large. Our theory predicts that as $V$ increases, the alignment between ZO and FO gradients diminishes. This directly explains the observation in [1] that naive ZO often fails on large-scale models.
>
> When we claim dimension-independent convergence, we specifically refer to the parameter dimension $d$ (the billions of weights), which has been the prohibitive bottleneck in classical ZO theory ($O(d)$). Our contribution shows that by shifting the dependency from $d$ to $V$, we explain why ZO can fine-tune billion-level models, provided the task-specific output space $V$ is manageable.
>
> ---
> > W2: Vision task.
>
> **Our response:** We clarify that the discrepancy in the required P mainly stems from the training regime (pre-training vs. fine-tuning) rather than differences between NLP and vision tasks. [2] studies pre-training from random initialization, where the eNTK is unstructured and isotropic, thus requiring a large P for accurate gradient estimation. In contrast, our work and [1] focus on fine-tuning a pre-trained model. As indicated by Eq. (17), the effectiveness of the ZO estimator depends on capturing informative directions. Pre-trained models exhibit a structured, low-rank eNTK, where gradients concentrate in a small subspace, allowing a much smaller P to suffice.
>
> ---
> > W3: About $\mu$.
>
> **Our response:**  We believe the reviewer is referring to the $O(\mu\eta)$ terms in Eq. (3). We use a first-order Taylor expansion to highlight the key mechanism while keeping the analysis intuitive. In fact, an exact characterization exists when perturbations are Gaussian (see [3], Sec. 2). Defining the Gaussian-smoothed objective
> $\ell_\mu(x) = \frac{1}{\kappa} \int \ell(x+\mu u)e^{-\|u\|^2/2} du$,
> the ZO estimator satisfies
> $E_u\left[ \frac{\ell(\theta+\mu u) - \ell(\theta)}{\mu} u \right] = \nabla \ell_\mu(\theta)$.
> The gap between $\ell_\mu$ and $\ell$, and between their gradients, can be bounded via higher-order $O(\mu^2)$ terms. We omit this more technical analysis for clarity.
>
> Regarding practice, $\mu$ is typically chosen as a small constant to control the $O(\mu^2)$ bias. While increasing $\eta$ can accelerate optimization, if it is too large, the $O(\mu\eta)$ terms become non-negligible and amplify gradient variance, degrading ZO performance. This explains why ZO works well in fine-tuning (small $\eta$) but is less effective in pre-training (large $\eta$).
>
> [3] Nesterov, Y. and Spokoiny, V. Random gradient-free minimization of convex functions. Foundations of Computational Mathematics, 2017
>
> ---
> > W4: hyperparameter setup and new SST-5.
>
> **Our response:**
> 1. For Figure 5, we set the learning rate $\eta \in \{1e-7, 1e-6, 5e-6, 1e-5\}$ and the smoothing radius $\mu = 1e-3$, following [1, 2].
> 2. FO vs. ZO-SGD (P=1): The close tracking of ZO-SGD with P=1 to FO-SGD is a key empirical finding. In the fine-tuning regime of LLMs, the loss landscape is effectively low-rank, so even a single random perturbation can capture dominant gradient directions with high probability. Although noisier, the update direction remains sufficiently aligned with FO-eNTK to yield similar optimization dynamics over time.
> 3. We conduct extra experiments on SST-5 (Table 1). To quantify trajectory differences, we compute the point-wise absolute difference between FO and ZO. The results show a consistent pattern with Figure 5 (SST-2): as P increases, ZO trajectories become closer to FO-SGD. We also observe that on SST-5, ZO with P=1 doesn't always outperform FO-SGD. Overall, the gap between ZO and FO decreases as P increases.
>
> Table 1: Mean Absolute Difference between ZO and First-Order SGD training accuracy trajectory on **SST-5**
>
> |Model| $P=1$  | $P=5$  | $P=10$ | $P=20$ | $P=50$ | $P=100$ |
> | --- | ---- | ---- | ---- | ---- | ---- | ---- |
> |125M |0.0333|0.0165|0.0172|0.0089|0.0108|0.0088|
> |350M |0.0274|0.0109|0.0109|0.0122|0.0087|0.0054|
> |1.3B |0.0861|0.0274|0.0296|0.0178|0.0113|0.0090|

---

> > ### Author Rebuttal · Reviewer_wBCZ · 2026-04-03
> >
> > Thank you for the rebuttal and for addressing W1 and W2. Regarding W3, I think it is not sound to use linear approximation in analyses. This paper claims "ZO methods can achieve a dimension-independent convergence rate even without the
> > strict low-effective-rank assumption on the Hessian", so it is essentially comparing to analyses in prior work such as [1,2]. However, these works do not use linear approximation. Instead, [2], for example, explicitly discusses the $\mu$ value to take to obtain $d$-independent convergence rate. To make this paper's claim hold under a fair comparison to other work, I think the assumption of linear approximation should not be made.
> >
> > [1] Fine-Tuning Language Models with Just Forward Passes, NeurIPS 2023.
> > [2] DPZero: Private Fine-Tuning of Language Models without Backpropagation. ICML 2024.

---

> > > ### Author Response · Authors · 2026-04-05
> > >
> > > We sincerely thank the reviewer for the continued engagement and for pointing out the important references [1,2]. We appreciate the opportunity to clarify the role of the linear approximation and its relation to prior work.
> > >
> > > First, we would like to clarify that the linear (first-order) approximation used in our paper is not a fundamental assumption required for our results, but rather an expositional and analytical tool to improve clarity and intuition. All results can be extended without relying on this approximation. Specifically, using the exact Gaussian smoothing formulation,
> > > $E_u \left[ \frac{\ell(\theta+\mu u) - \ell(\theta)}{\mu} u \right] = \nabla \ell_\mu(\theta)$,
> > > the same derivation can be carried out by replacing $\nabla \ell(\theta)$ with $\nabla \ell_\mu(\theta)$. Under standard smoothness assumptions, the difference between $\nabla \ell_\mu(\theta)$ and $\nabla \ell(\theta)$ is bounded by $\mathcal{O}(\mu^2)$. While this introduces additional technical complexity, it does not affect the qualitative conclusions of our analysis, including the dimension-independent behavior.
> > >
> > > Second, our use of linear approximation can be understood as analyzing the regime where $\mu$ is sufficiently small, which is also the standard setting in prior work. From a theoretical perspective, zeroth-order methods involve two distinct effects:
> > >
> > > - **Smoothing bias (controlled by $\mu$):**
> > >   The parameter $\mu$ determines the approximation error between the true objective and the smoothed objective. As analyzed in [2], this bias can be made arbitrarily small by choosing $\mu$ sufficiently small.
> > >
> > > - **Gradient variance (dimension-dependent):**
> > >   The primary bottleneck underlying the classical "dimension dependence" of ZO methods is the variance of the gradient estimator, rather than the smoothing bias.
> > >
> > > Our analysis focuses on separating these two effects. Since the bias can be controlled independently via $\mu$, adopting a first-order approximation (equivalently, considering the small-$\mu$ regime) provides a clean way to isolate the variance structure that governs the dimension dependence. In this sense, our treatment is consistent with the standard theoretical regime considered in [1,2], without requiring a specific scaling of $\mu$ with the model dimension.
> > >
> > > Third, we would like to emphasize that the primary contribution of this paper is the characterization of the learning dynamics of zeroth-order methods. The dimension-independent convergence behavior arises as a consequence of this analysis, rather than being the sole objective. In contrast, prior works such as [1,2] primarily analyze optimization error bounds with explicit bias–variance trade-offs. Our perspective is complementary: by analyzing the induced learning dynamics, we show that the key factor governing performance is the variance structure of the estimator. This also enables a more direct, apples-to-apples comparison under the same practical regime where $\mu$ is taken to be a sufficiently small constant.
> > >
> > > We thank the reviewer again for pointing this out and will revise the paper to make these clarifications more explicit.

---

### Decision · Program_Chairs · 2026-04-30

**Decision:**

Accept (regular)

**Comment:**

This paper analyzes the learning dynamics of zeroth-order optimization (ZO) from the perspective of the empirical neural tangent kernel (eNTK), with the goal of theoretically explaining the empirical observation that ZO methods can work effectively in large language model (LLM) fine-tuning despite the high dimensionality of model parameters.

The main contribution lies in reinterpreting the difficulty of convergence that depends on the parameter dimension in classical analyses, by viewing ZO through the lens of learning dynamics in function space. In particular, the paper shows that a ZO update can be understood as approximating a first-order update via a randomly projected low-dimensional version of the eNTK. This perspective reveals that the approximation quality of ZO depends primarily on the output dimension and the number of perturbations, rather than on the parameter dimension. This provides a new explanation for the practical success of ZO methods in LLM fine-tuning. This conceptual shift is regarded as novel and meaningful by multiple reviewers.

On the other hand, a key concern is that part of the theoretical analysis relies on a first-order Taylor approximation. Some reviewers maintained concerns about the validity of this approximation and the fairness of comparisons with prior work that does not rely on such assumptions. This issue should be more clearly positioned and clarified in the final version. In addition, the theory in this paper is fundamentally based on one-step update approximations and the lazy training regime, and therefore does not extend to settings such as training from scratch or tasks with strong sequential dependencies. While the authors clarified this scope limitation in the rebuttal, it would be desirable to state it more explicitly in the main text.

Taking these points into account, I believe the paper is worthy of acceptance, conditional on clarifying the role of the approximation and the underlying assumptions, and providing a more balanced discussion of its relationship to existing ZO theory.